# Data Quality Matters For Adversarial Training: An Empirical Study

## Abstract

Multiple intriguing problems are hovering in adversarial training, including robust overfitting, robustness overestimation, and robustness-accuracy trade-off. These problems pose great challenges to both reliable evaluation and practical deployment. Here, we empirically show that these problems share one common cause—low-quality samples in the dataset. Specifically, we first propose a strategy to measure the data quality based on the learning behaviors of the data during adversarial training and find that low-quality data may not be useful and even detrimental to the adversarial robustness. We then design controlled experiments to investigate the interconnections between data quality and problems in adversarial training. We find that when low-quality data is removed, robust overfitting and robustness overestimation can be largely alleviated; and robustness-accuracy trade-off becomes less significant. These observations not only verify our intuition about data quality but may also open new opportunities to advance adversarial training.

## 1 Introduction

Adversarial training (Goodfellow et al., 2015; Huang et al., 2015; Kurakin et al., 2017; Madry et al., 2018) is arguably the most effective way to establish the robustness of deep neural networks against adversarial perturbations. Nevertheless, several intriguing problems and phenomena hover in adversarial training practice, including (1) robust overfitting (Rice et al., 2020), (2) robustness overestimation[1] (Uesato et al., 2018; Mosbach et al., 2018; Croce & Hein, 2020b; Chen & Gu, 2020), and (3) robustness-accuracy trade-off (Papernot et al., 2016; Su et al., 2018; Tsipras et al., 2019; Zhang et al., 2019).

In this work, we focus on the data in adversarial training and show that the three aforementioned problems are all interconnected with the quality of the data employed in adversarial training. Specifically, we measure the data quality based on their stability of being learned through training and show that unstably learned examples are of low quality to adversarial training. We show that low-quality data can be one common cause of the three aforementioned problems. To the best of our knowledge, this is the first time the effect of the data quality being systematically studied in adversarially robust learning. This is also the first time that all these problems are investigated in a holistic manner.

As a demonstration, we partition the training set of CIFAR-10 (Krizhevsky, 2009) into two exclusive subsets with equal size and balanced classes, while ensuring all the examples in one subset have higher quality than all the examples in the other one in a class-wise manner. We then conduct both standard training and adversarial training with Projected Gradient Descent (PGD-AT) (Madry et al., 2018). The model is fixed as pre-activation ResNet-18 (He et al., 2016). As shown in Figure 1, adversarial training on the high-quality half yields *high* robustness, *almost no* robust overfitting, *genuine* robustness against the popular PGD attack (Madry et al., 2018), and *minor* robustness-accuracy trade-off that is only noticeable in the late stage of training, which is in sharp contrast to that on the low-quality half.

We further conduct controlled experiments on three real-world datasets, CIFAR-10, CIFAR-100 (Krizhevsky, 2009) and Tiny-ImageNet (Le & Yang, 2015), to explore the interconnection

---

[1]Robustness overestimation refers to the problem that the adversarial robustness may be spuriously high against certain types of adversaries.

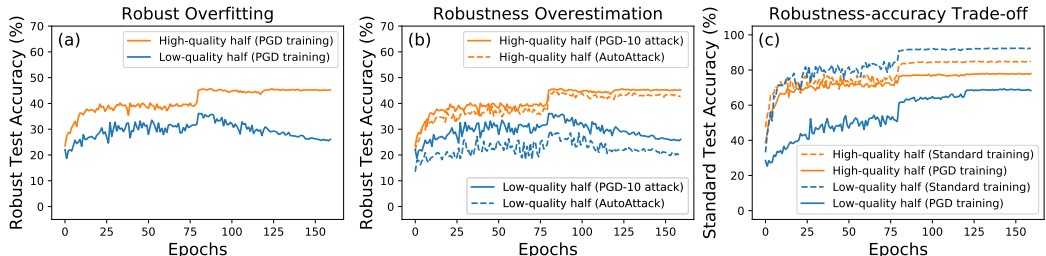

Figure 1: Despite being equal-size and class-balanced, the high-quality half and low-quality half of the CIFAR-10 training set have drastically different behaviors in adversarial training.

between data quality and those three problems in adversarial training. Our experiments cover various sample sizes, training methods (e.g., PGD-AT and TRADES (Zhang et al., 2019)), and neural architectures (e.g., pre-activation ResNet and Wide ResNet (WRN) (Zagoruyko & Komodakis, 2016)).

Our major findings are summarized as follows.

- Low-quality data may not be useful or even detrimental to adversarial training.
- Low-quality data causes robust overfitting in adversarial training.
- Adversarial training with low-quality data can lead to overestimated robustness against certain adversaries such as the PGD attack.
- Robustness-accuracy trade-off is only prominent when the dataset contains low-quality data. When using high-quality data only, the standard accuracy achieved by adversarial training and standard training is comparable.

The remainder of this paper is organized as follows. In Section 2, we briefly review the existing works that are most related to our data-centric investigation of adversarial training. In Section 3, we propose a strategy to estimate the data quality based on learning stability, and show that unstably-learned data are of low quality to adversarial training. Sections 4.1, 4.2 and 4.3 investigate the interconnection between data quality and major problems in adversarial training including robust overfitting, robustness overestimation, and robustness-accuracy trade-off respectively. Conclusions and further implications are discussed in Section 5.

## 2 RELATED WORK

Increasing attention has recently been paid to analyzing and improving the quality of the (training) data. Compared with standard learning, only a limited number of all learnable features would be beneficial to the adversarial robustness (Ilyas et al., 2019), thus making data quality more important in adversarially robust learning. Existing works have been focusing on the effect of sample size and data distribution in adversarial training. Schmidt et al. (2018) show that adversarially robust learning requires a much higher sample complexity than standard learning. They also show that such sample complexity requirement is highly sensitive to the data distribution. Ding et al. (2019) further observe that adversarial robustness achieved by adversarial training is sensitive to semantically-lossless distribution shift. Shafahi et al. (2019) show that the adversarial robustness achieved by any classifier is fundamentally limited by the properties of the data distribution. Our study is distinguished from previous works by investigating the datasets with the same size and within the same distribution. We also systematically study the other important problems in adversarial training beyond the performance.

Towards advancing adversarial robustness, substantially more works have been devoted to customizing the inner maximization or outer minimization in adversarial training based on the properties and behaviors of training data. Representative works including Wang et al. (2020), which shows that misclassified examples have greater impacts on adversarial robustness, and Zhang et al. (2020b), which observes that some training data are more robust to adversarial attack and will significantly improve the robustness against certain types of attacks if they are emphasized in adversarial training. Following a curriculum adversarial training framework (Cai et al., 2018), Wang et al. (2019) proposes to gradually increase the convergence quality of adversarial examples through training by observing that training on adversarial examples with high convergence quality in the later stages leads to better robustness. Selecting high-quality data may be more crucial when including extra data into training

that are either unlabeled (Uesato et al., 2019; Carmon et al., 2019; Gowal et al., 2020) or generated by generative models (Sehwag et al., 2021; Gowal et al., 2021), where the predictive probability of a model is comomly used to measure the data quality. We review more works on the different measures of data quality and their applications in Appendix D. Our study is distinguished from these works by both methodology and objective. In terms of the methodology, we identify the low-quality examples based on their behaviors of being learned throughout the adversarial training. Our approach to measure data quality yields more fine-grained results than misclassification to distinguish the individual examples, and more consistent results than the resistance to perturbation or predictive probability among different training methods and neural architectures (see Appendix D). In terms of the objective, we are not aiming to compete with the state-of-the-art methods, but rather to reveal the important influence of data quality on the problems in adversarial training through rigorously controlled experiments and reliable robustness evaluation. Such comparative and systematic analyses are lacking in the current adversarially robust learning research.

Due to the space constraint, we will discuss more works specifically related to the benefit of data in adversarially robust learning, robust overfitting, robustness overestimation, and robustness-accuracy trade-off when it comes to Sections 3.2, 4.1, 4.2 and 4.3, respectively. More broadly related work regarding data quality in standard and robust learning will be discussed in Appendix A.

## 3 DATA QUALITY IN ADVERSARIAL TRAINING

### 3.1 DATA QUALITY MEASURE: LEARNING STABILITY AS AN EXAMPLE

We propose to measure the quality of the data in adversarial training based on the frequency of events that an example is predicted correctly under adversarial perturbation throughout the training, which we will refer as the *learning stability* in the rest of the paper. Specifically, given an example $x$, the model $f$ and the optimizer $\omega$, the learning stability $s(x; f, \omega)$ can be formulated as

$$s(x; f, \omega) = \frac{1}{T} |\{t | f(t, x + \delta) = y(x), \ t \in \{1, \cdots, T\}\}|, \qquad (1)$$

where $\delta$ is a $\ell_\infty$ norm-bounded adversarial perturbation, $f(t, \cdot)$ denotes the classifier at epoch $t$, $y(x)$ denotes the true label of example $x$, and $T$ is the total number of epochs the model will be trained. A similar measure in standard training has been utilized by Moon et al. (2020) to regularize the confidence estimates of deep neural networks, and is shown to be roughly proportional to the probability that an example will be correctly predicted by a model.

We find that, although the absolute value of the learning stability can vary a lot (see Appendix C.2), the ranking of examples based on the learning stability is consistent across different experimental setups, such as random initializations, training epochs, training methods, and neural architectures as shown in Figure 2. This suggests that the learning stability can reflect the *intrinsic* "hardness" of an example in a *relative manner*.

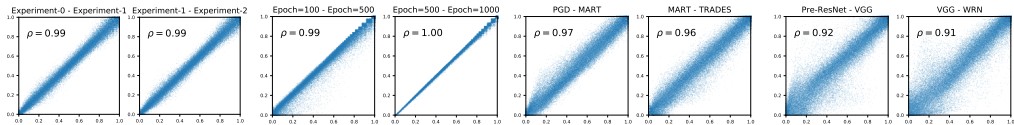

(a) Random initializations    (b) Training epochs    (c) Different methods    (d) Different models

Figure 2: Correlation between the rankings of the examples (scaled by the total number of examples) based on learning stability estimated by different training settings. Spearman's rank coefficient ($\rho$) is annotated at the upper left of each subfigure. Here we consider adversarial training methods including PGD-AT, TRADES and MART (Wang et al., 2020), and neural architectures including pre-activation ResNet-18, VGG-19 (Simonyan & Zisserman, 2015) and WRN-16-10.

Towards a more accurate estimation of such hardness, we calculate the ranking of the learning stability based on the ensemble of multiple experiments (see Appendix C.1). Figure 3a and Figure 3b show samples of the examples with the lowest learning stability and the examples with the highest learning stability, respectively, in the CIFAR-10 training set. One can find that unstably-learned examples appear more ambiguous and may not align well with their given labels. Under adversarial

| airplane | automobile | bird | frog | ship | airplane | automobile | bird | frog | ship |

| bird | ship | cat | ship | airplane | airplane | automobile | bird | frog | airplane |

(a) Examples with the lowest learning stability  (b) Examples with the highest learning stability

Figure 3: Examples selected from CIFAR-10 training set by learning stability. The top row shows the original images annotated with their true labels and the bottom row shows the corresponding adversarially-perturbed images annotated with their labels predicted by a model.

perturbation with relatively small size, they cannot easily be identified even by humans. This aligns with our intuition that unstably-learned examples may be intrinsically hard.

In standard learning, hard examples are often essential to the model performance, which is manifested in real-world applications such as hard example mining (Kumar et al., 2010; Shrivastava et al., 2016) and ablation studies on the effect of removing hard examples (Toneva et al., 2019). Empirical evidence can also be found in our Figure 1(c) where standard training on the unstably-learned half yields a higher standard accuracy than that on the stably-learned half.

In adversarial training, rather surprisingly, we observe an opposite case. Unstably-learned examples are often not useful and may even hurt the robustness achieved by adversarial training (see Section 3.2). Therefore, in the rest of the paper, we refer those unstably-learned examples as *low-quality examples*, and measure the data quality relatively using the learning stability. It is worth mentioning that, in Appendix D, we show that it is possible to estimate the data quality based on other behaviors and properties of the data (e.g., predictive probability, minimum perturbation, and learning order) and all these estimations will yield consistent findings.

## 3.2 UNSTABLY-LEARNED DATA IS OF LOW-QUALITY FOR ADVERSARIAL TRAINING

In this section, we conduct controlled experiments on three real-world datsets and show that low-quality examples are not useful and may even be detrimental to adversarial training.

**Related work.** In adversarially robust learning, solely increasing sample size will not always yield better performance and sometimes even be detrimental. Ding et al. (2019) observes that the robustness achieved by adversarial training plateaus after training size is sufficiently large on MNIST (LeCun et al., 1998). Introducing additional data examples may not always help the adversarial robustness if they are unrelated to the original dataset (Uesato et al., 2019; Gowal et al., 2020). Yang et al. (2020a) shows that a proper pruning of the dataset will benefit the adversarial robustness for non-parametric classifiers. A recent study also shows that adversarial vulnerability may originate from the label noise in the dataset (Sanyal et al., 2020), which apparently hurts the adversarial robustness. However, we note that low-quality data is not simply label noise after manually inspecting the samples as those in Figure 3a. Moreover, in-depth studies show that the fractions of label noise in benchmark datasets are only marginal (Northcutt et al., 2021).

**Experimental setup of controlled experiments.** We conduct controlled experiments to dissect the effect of training data on adversarial training. Specifically, we gradually remove more examples from the training set and record the robustness achieved by adversarial training from scratch on an increasingly smaller training subset, while all other experiment settings remain the same. The training examples are removed either randomly, or in an ascending order of data quality, i.e., low-quality data will be removed first. By removing the same number of random data and low-quality data, it should be able to eliminate the effect of sample size in the comparison and reveal the effect of data quality.

We consider the robustness against $\ell_\infty$ norm-bounded adversarial attack with perturbation radius 8/255 throughout the paper. We evaluate the adversarial robustness against AutoAttack (AA) (Croce & Hein, 2020b), one of the adversarial attacks known to be relatively reliable. We employ popular adversarial training methods including PGD-AT and TRADES combined with early stopping (Rice et al., 2020), which have been shown to attain the improvements of almost all other adversarial training

variants (Croce & Hein, 2020b; Chen & Gu, 2020; Pang et al., 2021). We conduct experiments with pre-activation ResNet-18 in the main paper and Wide ResNet in the appendix. We present results on three datasets including CIFAR-10, CIFAR-100 and Tiny-ImageNet. Other experiment details can be found in Appendix F. The same experimental setup will be applied to Sections 4, 5 and 6 as well.

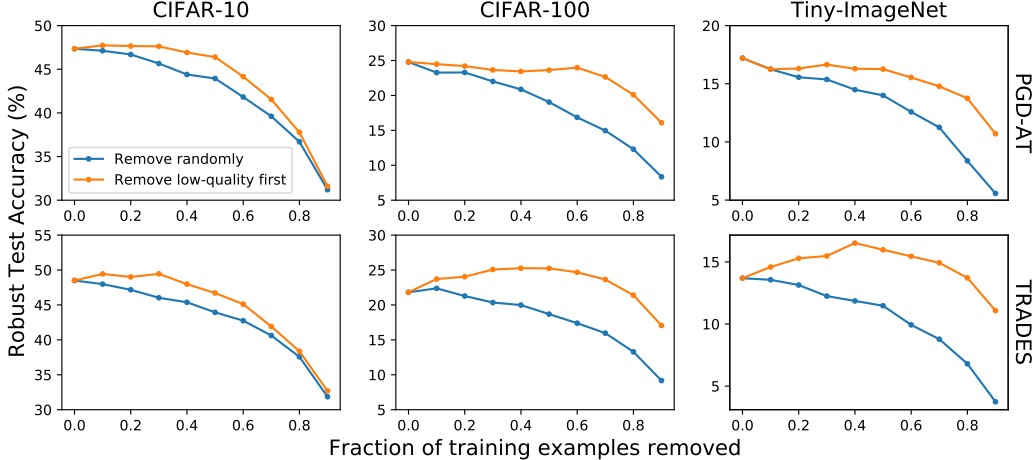

Figure 4: Best robust test accuracy obtained when removing different fractions of training examples either randomly or in an ascending order of data quality.

**Results & Discussions.** Figure 4 shows that removing random examples from the training set degrades the robustness achieved by adversarial training rapidly. In contrast, when removing low-quality examples first, the achieved robustness maintains or even increases. For instance, by PGD-AT, up to $40\%$, $60\%$, and $50\%$ low-quality training examples can be removed without reducing the robustness significantly $(< 1\%)$ on CIFAR-10, CIFAR-100, and Tiny-ImageNet, respectively. More surprisingly, removing examples with the lowest data quality can achieve robustness even higher than that achieved on the entire training set for both PGD-AT and TRADES on CIFAR-10, and TRADES for other datasets, although the training set is actually becoming smaller. We also validate the above observations with different perturbation sizes and neural achitectures in Appendix B.2.

## 4 DATA QUALITY AND PROBLEMS IN ADVERSARIAL TRAINING

In this section, we investigate the interconnection between data quality and problems in adversarial training by conducting extensive controlled experiments.

### 4.1 ROBUST OVERFITTING

**The problem and related work.** *Robust overfitting* is a prevalent phenomenon in adversarial training (Rice et al., 2020). Specifically, the robust test accuracy will constantly decrease after a certain point in adversarial training, resulting in an inferior final performance. Such overfitting occurs consistently across different datasets, training settings, adversary settings, and neural architectures, and cannot be completely eliminated other than using early stopping (Rice et al., 2020).

**Low-quality data causes robust overfitting.** Our controlled experiments suggest that the robust overfitting results from the low-quality examples. Figure 1 already shows that adversarial training on the high-quality half induces almost no robust overfitting. This suggests that robust overfitting is against our conventional understanding of overfitting in the sense that it can be mitigated by a smaller training sample size. Figure 5 further shows that when a particular fraction of low-quality examples is removed from the training set (e.g., on CIFAR-10, about $30\%$ for PGD-AT and $50\%$ for TRADES), the robust overfitting would vanish. The smaller number of low-quality examples are removed from the training set, the more severe the robust overfitting will be. In contrast, when the examples are randomly removed from the training set, the robust overfitting is often prominent consistently across different training set sizes. This suggests that low-quality examples cause the robust overfitting, and more low-quality examples will make it more severe.

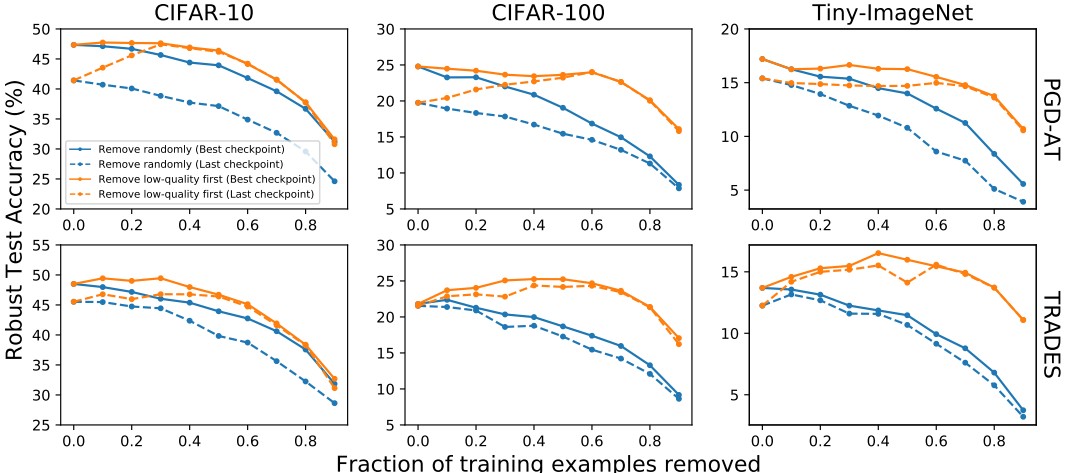

Figure 5: Robust test accuracy obtained at the best and last checkpoints by adversarial training when removing training examples either randomly or in an ascending order of data quality. The robust overfitting can be captured by the visual gap between the dashed and solid curves of the same color.

Motivated from the feature learning perspective, we conjecture that low-quality examples cause robust overfitting because robust features might be forgotten when low-quality examples are adversarially learned. More detailed explanation can be found in Appendix G.2.

**Mitigate robust overfitting.** Based on the above understanding, to mitigate robust overfitting, one may need to intentionally prevent the model from learning low-quality examples. Removing low-quality examples from the training set and the early stopping (Rice et al., 2020) are the most straightforward methods to prevent such learning. Recent works to mitigate robust overfitting including regularizations of the flatness of the weight loss landscape (Wu et al., 2020; Stutz et al., 2021), introducing low-curvature activation functions (Singla et al., 2021) and adopting stochastic weight averaging (Izmailov et al., 2018) and knowledge distillation (Hinton et al., 2015) (Chen et al., 2021). These methods are likely to be consistent with our data-centric understanding as various regularization techniques may suppress the learning of low-quality examples.

## 4.2 ROBUSTNESS OVERESTIMATION

**The problem and related work.** We refer *robustness overestimation* as the problem that the robustness of an adversarial training method may appear spuriously high against certain types of adversaries, but be significantly diminished by stronger adversaries (Chen & Gu, 2020). Such problem might result from a significantly more difficult inner maximization induced by highly non-linear loss landscape (Qin et al., 2019), thus is more subtle than the conventional *gradient masking* problem where the gradients of the model are completely not useful (Papernot et al., 2017; Tramèr et al., 2018; Athalye et al., 2018; Uesato et al., 2018; Engstrom et al., 2018).

We focus on the PGD attack as it is the most popular method to evaluate adversarial robustness. Previous works have already shown that PGD attack with insufficient iterations (Mosbach et al., 2018) and fixed step size (Croce & Hein, 2020b) may yield misleading robustness. Existing adversarial training methods may suffer from unreliable evaluation against PGD attack as mentioned in recent comprehensive studies (Croce & Hein, 2020b; Chen & Gu, 2020; Pang et al., 2021).

**Low-quality data can cause robustness overestimation.** For adversarial training methods, we show that low-quality data can cause overestimated robustness if evaluated against PGD attack by our controlled experiments. We report the **best robust test accuracy** throughout the training to rule out the effect of robust overfitting. We employ AutoAttack as a stronger adversary to evaluate the adversarial robustness. Although AutoAttack not necessarily reflects the true robustness (Tramèr et al., 2020), it is more reliable than PGD attack and is sufficient to reveal the significance of overestimation.

We denote the *overestimation gap* as the difference between the robust test accuracy evaluated against PGD-10 attack and AutoAttack. Figure 6 shows that the overestimation gap shrinks significantly when low-quality data is removed from the training set. For PGD-AT on all datasets, the robustness

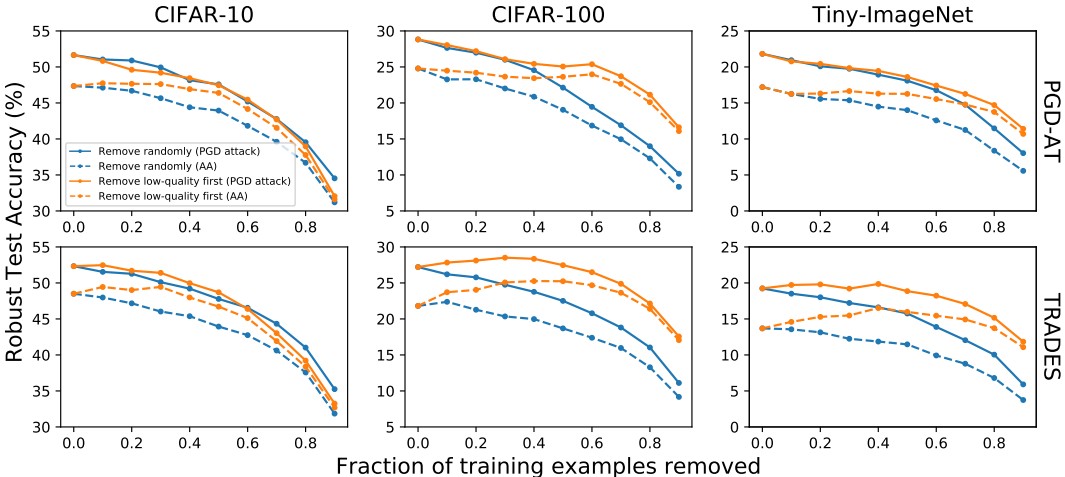

Figure 6: Best robust test accuracy obtained and evaluated against PGD attack and AA when removing training examples either randomly or in an ascending order of data quality. The overestimation gap can be captured by the visual gap between the dashed and solid curves of the same color.

obtained on the entire training set is often higher than that obtained on a smaller training set with low-quality data removed if evaluated against PGD attack, but will instead be close to or even lower than the latter if evaluated against AutoAttack. In contrast, when the data is removed randomly from the training set, the overestimation gap largely maintains and is also consistently larger than that produced by removing an equal amount of low-quality data. These evidences show that low-quality data in the training set is an important source of the robustness overestimation.

In Appendix B.3, we evaluate the robustness against a variety of black-box attacks including Square Attack (Andriushchenko et al., 2020), RayS (Chen & Gu, 2020) and transfer attack from a separately trained model, and show a consistent correlation between data quality and robustness overestimation.

From a geometrical perspective, we suspect low-quality data causes robustness overestimation as they lie close to the decision boundary (see Appendix D.2). The perturbation balls of low-quality data from different classes may overlap, thus twisting the local loss landscape and complicating the maximization problem in gradient-based attacks (See more discussion in Appendix G.3).

Table 1: Robust test accuracy achieved by various training methods on the entire training set, evaluated by various attack methods. "*" indicates the transfer attack from a surrogate model.

| | White-box Attack | | | Black-box Attack | | |
| Dataset | PGD-10 (%) | PGD-1000 (%) | AA (%) | Square (%) | RayS (%) | PGD-1000* (%) |
|---|---|---|---|---|---|---|
| PGD | 52.07 | 51.04 | 48.12 | 56.19 | 56.86 | 62.24 |
| GAIRAT | 59.51 | 59.33 | 32.34 | 41.98 | 44.17 | 58.06 |
| Diff | +7.44 | +8.29 | −15.78 | −14.21 | −12.69 | −4.18 |
| TRADES | 52.56 | 51.75 | 48.58 | 55.20 | 55.63 | 63.19 |
| MART | 53.81 | 53.39 | 46.57 | 52.76 | 53.45 | 60.73 |
| Diff | +1.25 | +1.64 | −2.01 | −2.44 | −2.18 | −2.46 |

**Implications.** Since low-quality data can cause robustness overestimation, it can be intentionally exploited to create spuriously high robustness against weak adversaries such as PGD attack. We find that this problem is ubiquitous among sophisticated adversarial training variants. We select two recently published methods GAIRAT (Zhang et al., 2020b) and MART and show that they suffer from robust overestimation due to the emphases on low-quality data in their training objectives (See Appendix A.1). As shown in Table 1, these methods trained with recommended settings (see Appendix F) can achieve significant improvements compared to their corresponding baselines when evaluated against white-box PGD attack, even with a large number of attack iterations. However, when evaluating against a stronger adversary such as AutoAttack, or black-box attacks, the performance of these methods are instead inferior. Similar observations have been made for these methods in recent works (Gowal et al., 2020; Hitaj et al., 2021). These results imply that emphasizing on the

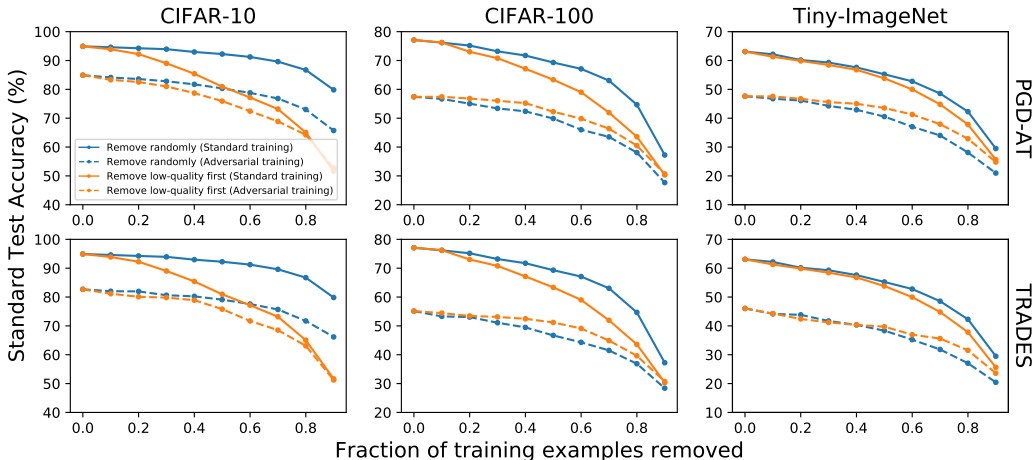

Figure 7: Standard test accuracy obtained by standard training and adversarial training when removing different fractions of training examples either randomly or in ascending order of data quality. The cross-generalization gap can be captured by the visual gap between the dashed and solid curves of the same color.

low-quality data in adversarial training may not be appropriate and it is important to validate the robustness against a wide variety of evaluation metrics as more methods are focusing on customizing the training for individual examples.

### 4.3 ROBUSTNESS-ACCURACY TRADE-OFF

**The problem and related work.** It is widely observed that adversarially training the model comes at the cost of standard accuracy, which is typically referred as the *robustness-accuracy trade-off* (Papernot et al., 2016; Su et al., 2018; Tsipras et al., 2019; Zhang et al., 2019). For certain learning problems, this trade-off might be inevitable either because no optimal classifier exists (Tsipras et al., 2019; Zhang et al., 2019) or the hypothesis classifier is not expressive enough (Nakkiran, 2019). Meanwhile, this trade-off is data-sensitive. Specifically, Ding et al. (2019) showed that the level of trade-off depends on the data distribution, and Tsipras et al. (2019) observed that the adversarial training is beneficial to standard accuracy when the training data is insufficient to train the model.

**Robustness-accuracy trade-off is correlated with the data quality.** We show that the robustness-accuracy trade-off is sensitive to the data quality within the same dataset. On high-quality data, this trade-off is not significant; however, on low-quality data, it is prominent.

We present the correlation between data quality and *cross-generalization gap* by controlled experiments. Following previous work (Chen et al., 2020), we define the cross-generalization gap as the standard test accuracy difference between standard training and adversarial training on the same dataset, which reflects the degree of robustness-accuracy trade-off induced by adversarial training on a specific dataset. Note here we report the standard test accuracy **at the last checkpoint** for all settings in case the selected checkpoint varies due to robust overfitting. As in Figure 7, the cross-generalization gap maintains the same or even enlarges as more randomly selected examples are removed from the training set. In contrast, the cross-generalization gap gradually shrinks as more low-quality examples are removed from the training set. For example, on all three datasets, adversarial training (either PGD-AT or TRADES) and standard training yield comparable standard accuracy, when using the 10% highest quality examples only (i.e., 90% lowest quality ones removed).

**Implications.** Our identified correlation between data quality and robustness-accuracy trade-off complements the existing observations of the impact of data distribution on robustness. Besides the training set size (Tsipras et al., 2019) and distribution shift (Ding et al., 2019), the quality of the training data can also influence this trade-off. In Figure 7, the cross-generalization gap induced by PGD-AT is negative when using the 10% most high-quality examples on CIFAR-10 and CIFAR-100, which means that adversarial training is actually beneficial to the standard accuracy. Similar phenomena were previously only observed on MNIST or extremely small training sets

($\sim 100$ examples) (Tsipras et al., 2019). Meanwhile, low-quality examples dominate the trade-off potentially because they are intrinsically ambiguous as shown in Figure 3a, where robust features may be difficult to learn. This is aligned with the understanding of trade-off from the feature learning perspective (Tsipras et al., 2019; Ilyas et al., 2019). More detailed explanations motivated by this perspective will be discussed in Appendix G.1.

**Mitigate robustness-accuracy trade-off.** Based on the above understanding, to mitigate the robustness-accuracy trade-off one may resort to properly deal with low-quality data in adversarial training. Multiple existing works are likely to implicitly perform this by customizing either the inner maximization (Balaji et al., 2019; Ding et al., 2020; Cheng et al., 2020; Zhang et al., 2020a) or outer minimization (Huang et al., 2020; Wang et al., 2020; Zhang et al., 2020b) for individual examples (see more in Appendix D). Although these methods can mitigate the trade-off by increasing either robustness or accuracy while maintaining the other, the effectiveness may be unclear under state-of-the-art robust evaluation methods (Croce & Hein, 2020b) as shown in Section 4.2.

## 5 CONCLUSION AND DISCUSSIONS

In this work we systematically study the difficulties in adversarial training from a data quality perspective. We show that the data quality is an important but previously neglected aspect for adversarial training. It is strongly correlated with the current problems in adversarial training including robust overfitting, robustness overestimation and robustness-accuracy trade-off.

Towards understanding the difficulty of robust learning, our analyses of the data quality complements the existing analyses on the impact of data properties such as distribution and sample size. Robust learning through adversarial training can exhibit diverse levels of difficulty on individual examples in the same benchmark dataset. From human perspective, low-quality data is often ambiguous, which implies that adversarial training might be susceptible to complex and indistinct image conditions, thus challenging the feasibility of adversarial training in a realistic scenario where ambiguous data may prevail. One may note that achieving robustness through adversarial training on ImageNet (Russakovsky et al., 2015) is already shown to be much harder (Rozsa et al., 2016; Kurakin et al., 2017).

In practice, our analyses of the data quality provides a new methodology to analyze the effectiveness of adversarial training methods. The learning behavior of a method on low-quality data reveals the potential reason why it may work or not work. For example, TRADES employs an alternative adversarial loss that penalizes the difference between the predictive probability from an adversarial example and its clean counterpart, instead of the true label, which will impose weaker supervision on low-quality examples since low-quality examples typically have low predictive probabilities (see Appendix D.1). As a result, TRADES can underfit low-quality data thus potentially gaining robustness compared to PGD-AT under the same perturbation setting, which resembles the effect of directly removing low-quality data.

Furthermore, our analyses encourage a new perspective to utilize the benchmark dataset for adversarial training. The fact that removing low-quality data maintains or even improves robustness implies that adversarial training cannot achieve the optimal robustness on the entire dataset, especially when the model capacity is relatively low. A promising direction to boost the robustness is to design methods able to learn low-quality data properly such that it can also contribute to the robustness. Nevertheless, We empirically find that no existing adversarial training method, without resort to larger model capacity, can achieve reliable robustness improvement through proper learning of low-quality data. Whether or not such a method exists remains an interesting problem.

Lastly, as it becomes increasingly popular to introduce additional data into adversarial training either labeled, unlabeled or generated (Hendrycks et al., 2019a; Uesato et al., 2019; Carmon et al., 2019; Najafi et al., 2019; Sehwag et al., 2021; Gowal et al., 2021), extra effort needs to be paid to picking high-quality data. The design of an effective and efficient data quality measurement is an important future work to excel in the performance.

**Limitations.** We emphasize that removing low-quality examples only serves as a demonstration of our intuition of data quality in adversarial training, and should not be advocated as a competing method. Although removing low-quality examples can potentially advance the best robustness, it will inevitably cost standard accuracy (see Appendix E), since those hard examples are of high quality to standard training.

## REPRODUCIBLITY STATEMENT

We conduct experiments on public benchmark. We will release implementations for all methods and scripts for all experiments on GitHub, under the Apache-2.0 license.

## ETHIC STATEMENT

In this paper, we conduct a series of controlled experiments to investigate the interconnections between data quality and problems in adversarial training. We experiment on three benchmark datasets that are publicly available. We have shown that low-quality examples are helpful for standard training but may hurt the adversarial robustness. We emphasize that removing low-quality examples only serves as a probing method to demonstrate our intuition of data quality in adversarial training, and should not be advocated as a competing method. Therefore, we believe our work is ethically on the right side of spectrum.

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

# A  MORE RELATED WORK

## A.1  ROBUSTNESS OVERESTIMATION

We discuss two recently published methods where the emphases on problematic data can be easily identified from their training objectives (See Table 2) compared to the baseline methods.

- GAIRAT (Zhang et al., 2020b) is based upon PGD-AT with an additional sample-wise weight in the loss function. Examples that are farther from the decision boundary will be assigned with larger weights, which amounts to an emphasis on the problematic data since the problematic score can be alternatively characterized by the distance to the decision boundary (see Appendix D.2).

- MART (Wang et al., 2020) can be largely viewed as a variant of TRADES with an additional term promoting the loss of those examples with low output probabilities, which also amounts to an emphasis on the problematic data based on the probability characterization of the problematic score (see Appendix D.1).

Table 2: Training objectives in the outer minimization in various adversarial training methods, where $\ell(\cdot, \cdot)$ indicates the cross-entropy loss, $f(\cdot)$ indicates the probabilistic prediction of a model and $f_y(\cdot)$ indicates the probability corresponding to class $y$, $\delta$ indicates the adversarial perturbation of $x$ generated by the inner maximization.

| Method | Training objective |
|---|---|
| PGD-AT | $\min_f \mathbb{E}_{x,y}\left[\ell(f(x+\delta), y)\right]$ |
| GAIRAT | $\min_f \mathbb{E}_{x,y}\left[\boldsymbol{\omega(x,y)}\ell(f(x+\delta), y)\right]$ |
| TRADES | $\min_f \mathbb{E}_{x,y}\left[\ell(f(x), y) + \lambda \cdot \ell(f(x), f(x+\delta))\right]$ |
| MART | $\min_f \mathbb{E}_{x,y}\left[\ell(f(x+\delta), y) + \lambda \cdot \ell(f(x), f(x+\delta))\boldsymbol{(1 - f_y(x))}\right]$ |

## A.2  ROBUSTNESS-ACCURACY TRADE-OFF

We note that there exists an abundant body of works that focuses on the robustness-accuracy trade-off in robust learning. In additional to the related works discussed in the main paper, here we review some works that attack this problem from other perspectives. This is by no means an exhaustive review.

It has been argued that the robustness-accuracy trade-off is inherent to the data distribution and is thus inevitable for any classifier. Tsipras et al. (2018; 2019) and Zhang et al. (2019) theoretically show that no optimal classifier can achieve both robustness and accuracy on toy problems. Dohmatob (2018) formalizes this into a "No Free Lunch" problem, and further proves the inevitability of trade-off under mild assumptions of the data distribution. Nakkiran (2019) shows that the trade-off is inevitable because the hypothesis class is not expressive enough. Javanmard et al. (2020) shows that the adversarial training may improve generalization in an over-parameterized regime, but hurt it in under-parameterized regime.

On the contrary, some works argue that the robustness-accuracy trade-off is not necessarily inevitable in a realistic setting. Raghunathan et al. (2020) shows that the trade-off stems from the over-parameterization of the hypothesis class. Robust self-training (Carmon et al., 2019; Najafi et al., 2019; Uesato et al., 2019), overcoming the sample complexity leveraging additional unlabeled data, thus can effectively mitigate the trade-off. Yang et al. (2020b) shows that the trade-off in practice is a result of either the model failing to impose local Lipschiztness, or not generalizing sufficiently. Wen et al. (2020) and Roth et al. (2020) show that adversarial training is essentially a form of operator norm regularization, thus hurting the generalizability if not properly configured.

There are more works implying that the robustness and accuracy may not be contradict by showing that adversarial examples can benefit generalization either through different perturbation generation strategies (Stutz et al., 2019) or different adversarial training strategies (Xie et al., 2020).

### A.3 DATA PROFILING IN STANDARD LEARNING

In classical machine learning, data quality is important because algorithms may be sensitive to noise and outliers. By measuring the degree of class overlapping and skewness in a dataset, Smith et al. (2013) proposes a generic definition of instance hardness representing how likely an example will be misclassified. Prudêncio et al. (2015) motivates from item response theory (IRT) (Embretson & Reise, 2000; Ayala, 2008) to characterize instance hardness. Smith & Martinez (2011) shows that properly removing hard examples in the dataset improves performance for a variety of learning algorithms including but not limited to Decision Tree and Support Vector Machine.

In deep learning regime, models with large capacity are typically more robust to outliers. Nevertheless, data examples can still exhibit diverse levels of difficulties. Arpit et al. (2017) finds that data examples are not learned equally when injecting noisy data into training. Toneva et al. (2019) shows that certain examples are forgotten frequently during training, which means that they can be first classified correctly then incorrectly. Model performance can be largely maintained when removing those least forgettable examples from training. Zhou et al. (2020) proposes to dynamically estimate instance hardness during training and encourage the model to focus on those hard examples from a curriculum learning (Bengio et al., 2009) perspective, which can improve both the performance and efficiency for a wide range of datasets and neural architectures. More generally, under a self-paced learning (Kumar et al., 2010) framework, diverse methods have been proposed to mine hard examples on the fly (Chang et al., 2017).

### A.4 DATA PROFILING IN ROBUST LEARNING

In robust learning regime, the model is required to learn features that are robust to perturbations. Such task is generally more difficult, where the data examples are thus more likely to differentiate in terms of their behaviours in learning or contribution to the model performance. Plenty of works have analyzed the diverse behaviours of data during adversarial training, and proposed a variety of methods that treating the data examples differently. A detailed review has been made in Section 2, and Section 4.3, 4.1, 4.2 that focus on existing problems in adversarial training specifically.

Here we mainly review works that investigate adversarial examples from a data perspective. Hendrycks et al. (2019b) collects a set of unperturbed images, known as the "natural adversarial examples", that significantly degrades the performance of state-of-the-art image classifiers. Pestana et al. (2020a) observes that the adversarial perturbations concentrate in the Y-channel of the YCbCr space, which is believed to contain more shape and texture related information. Pestana et al. (2020b) reports the existence of a set of images that are particularly robust to adversarial perturbations. Including such data in validation significantly limits the reliability of the robustness evaluation.

## B MORE EXPERIMENT RESULTS

### B.1 ADDITIONAL NEURAL ARCHITECTURES

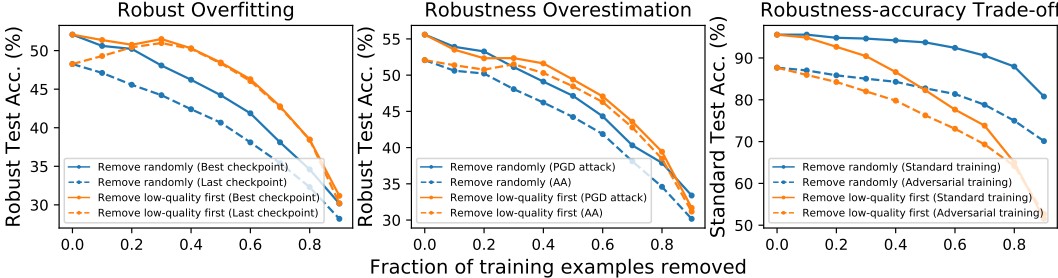

Figure 8: Correlation between data quality and three problems in adversarial training shown by conducting controlled experiments on WRN-28-10.

In this section we show the interconnection between data quality and those three problems in adversarial training including robust overfitting, robustness overestimation and robustness-accuracy trade-off also holds true for other neural architectures such as WRN. Here we conduct experiments on

WRN-28-10 and employ PGD training on CIFAR-10 as an example. As shown in Figure 8, as low-quality examples are removed from the training set, the robust overfitting would vanish, robustness overestimation and robustness-accuracy trade-off would become less significant, in contrast to the observations when removing randomly selected examples.

### B.2 DATA QUALITY AND ROBUSTNESS

**Experiments on different models and perturbation sizes.** For larger perturbation radii and smaller models, removing a certain group of low-quality examples yields more robustness improvement as shown in Figure 9a and Figure 9b respectively. Note that in Figure 9b, the robustness improvement yielded by removing 20% examples with the lowest quality diminishes as the model capacity increases. For considerably large models such as WRN-34-10 it will eventually hurts. These analyses match our intuition that adversarial examples produced by large perturbation are hard to learn; and smaller models are hard to learn adversarial robustness.

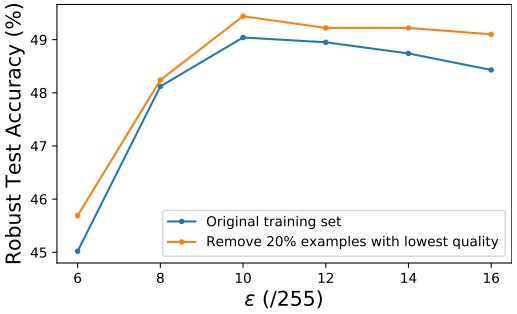
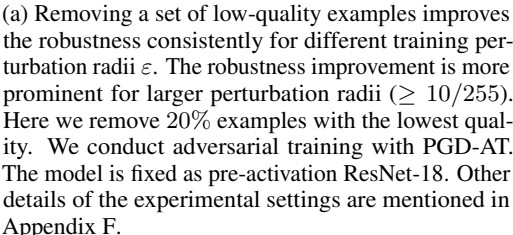
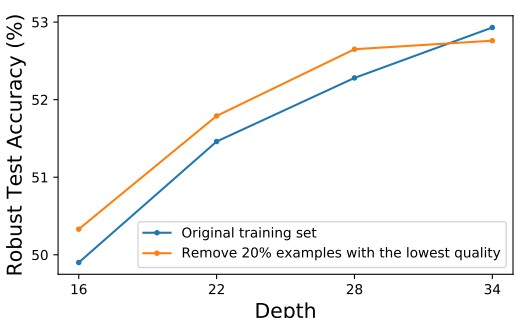

(a) Removing a set of low-quality examples improves the robustness consistently for different training perturbation radii $\varepsilon$. The robustness improvement is more prominent for larger perturbation radii ($\geq 10/255$). Here we remove 20% examples with the lowest quality. We conduct adversarial training with PGD-AT. The model is fixed as pre-activation ResNet-18. Other details of the experimental settings are mentioned in Appendix F.

(b) The robustness improvement by removing a set of low-quality examples is more prominent for smaller models, and gradually vanishes as the model capacity increases. Here we control the model capacity by modulating the depth of Wide ResNet, where the width is fixed as 10. We experiment on PGD-AT with the training perturbation radius yielding the best robustness ($12/255$).

### B.3 DATA QUALITY AND ROBUST OVERESTIMATION

**Experiments on more robustness evaluation methods.** To exclude the possibility that AutoAttack itself is sensitive to the data property due to its sophisticated design, we take CIFAR-10 as an example and select 20% as a representative fraction of low-quality examples being removed where the overestimation gap shrinks significantly. We further evaluate the robustness obtained by removing 20% low-quality examples against a variety of black-box attacks including Square Attack, RayS and transfer attack from a separately trained surrogate model (See Appendix F for details). We conduct repeated experiments (5 times) to reduce the statistical bias. We find that compared to the training set with 20% low-quality examples removed, the robustness obtained on the entire training set can be significantly higher ($\sim 2\%$) when evaluated against PGD-10 and PGD-1000, i.e. PGD attack with up to 1000 iterations. However, the improvement is actually not clear if evaluated against Square Attack and RayS in that the average difference is less than the standard deviation, and may even be negative for AutoAttack and transfer attack. In contrast, compared to the training set with 20% randomly selected examples removed, the robustness obtained on the entire training set is consistently higher for all evaluation attacks. This demonstrates that the low-quality data can cause difficulty for PGD attack to reliably evaluate the robustness, which cannot be avoided even with a substantially large number of iterations. This also implies PGD attack is likely to be suboptimal with fixed parameters (Mosbach et al., 2018; Croce & Hein, 2020b).

## C    MORE ABOUT DATA QUALITY

### C.1    CALCULATION OF THE DATA QUALITY RANK

To output a relatively accurate estimation of the data quality rank, we synthesize the results of multiple experiments. Specifically, we adversarially train a pre-activation ResNet-18 using PGD-10 on the entire CIFAR-10 training set for 160 epochs with learning rate initialized as 0.1 and decayed at epoch 80 and 120 with a factor of 10. We repeat the training 10 times and average the data quality ranks calculated on each example, which yields a relatively stable estimation[2].

### C.2    DISTRIBUTION OF LEARNING STABILITY

In Section 3.2, we estimate the data quality relatively by calculating the ranking of the examples based on learning stability. Directly using learning instability to measure the data quality might be troublesome as its magnitude varies greatly across different training settings (e.g. number of training epochs).

Figure 10 shows the distribution of the learning stability with different training epochs. One may observe that, the overall magnitude of the learning stability varies greatly as the training setting changes, which is reasonable since the model complexity hinges on the training settings and all training examples will eventually be overfitted given sufficient training epochs.

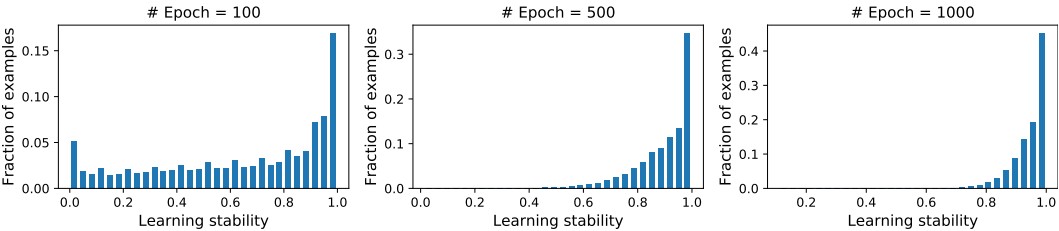

Figure 10: The distribution of learning stability given different training epochs.

## D    MORE METHODS TO MEASURE THE DATA QUALITY

In this section, we show that it is possible to estimate the data quality motivated from multiple measurements including prediction probability, minimum perturbation and learning order. Each measurement is itself consistent across different training settings, and is also correlated with the data quality rank estimated in Section 3.1. We also briefly review existing adversarial training methods leveraging these measurements to customize the inner maximization or outer minimization for individual examples in their designs, which implies that their improved performance is potentially due to treating examples of different data quality differently.

### D.1    PREDICTION PROBABILITY

Based on the standard adversarial training (Madry et al., 2018), multiple variants pivot on the utilization of soft output in the adversarial loss function. Here we specifically refer the soft output as the either the output before the softmax function, namely logit, or that after the softmax function, namely prediction probability. Adversarial logit pairing (ALP) (Kannan et al., 2018) is a method explicitly penalizing the difference between the logits from an clean example and its adversarial counterpart on top of the standard cross-entropy loss for adversarial training. BGAT (Cui et al., 2020) instead collects the logits of clean examples from an auxiliary clean model. GAT (Sriramanan et al., 2020) adopts a similar loss function penalizing probability difference instead of logit difference.

---

[2]Due to computational constraints, this estimation process only provides about 1600 unique values for data quality ranks, which are not enough to differentiate all the $5 \times 10^4$ training examples in CIFAR-10. However, in above analyses, we at most partition the training set into 10 subsets with different quality levels, which only results in about $0.6\%$ indistinguishable examples between any two adjacent subsets. One can seek more accurate estimation of data quality rank by incorporating the results from more experiments.

VAT (Miyato et al., 2019) and TRADES (Zhang et al., 2019) propose a loss function matching the probability from an adversarial example with its clean counterpart, instead of the true label. Self-adaptive training (Huang et al., 2020) and MART (Wang et al., 2020) use the probabilities from clean examples to weight the examples in the loss function, although the former focuses more on examples with high probabilities while the latter focuses more on examples with low probabilities.

Prediction probability is also related to whether an example is correctly classified or not. Low probability of the true label indicates an example is likely to be misclassified, which might be troublesome in adversarial training because adversarial examples of misclassfied examples are "undefined" (Wang et al., 2020). Several methods are thus motivated to treat correctly classified and misclassified examples differently. MMA (Ding et al., 2020) employs adverarial training only on correctly classified examples, leaving misclassified examples to standard training. MART (Wang et al., 2020) introduces a term associated with the probability of the true label in the loss function to encourage the learning on misclassified examples. A summary of the loss functions employed in these methods can be found in Table 1 by Wang et al. (2020).

Recently, adversarial training with additional data becomes increasingly popular. Prediction probability is used to identify high-quality data that is more relevant to the original distribution (Gowal et al., 2020).

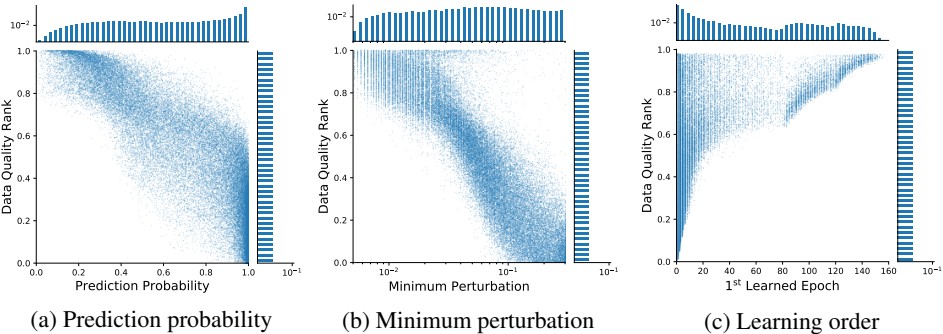

(a) Prediction probability  (b) Minimum perturbation  (c) Learning order

Figure 11: Correlation between data quality rank and other measurements

Here we show that the prediction probability of an example is correlated with its data quality rank. These sophisticated methods are thus likely to achieve robustness gain by treating examples with different quality levels differently. We specifically refer the *prediction probability* as the probability corresponding to the true label from a clean input, and use the best model in terms of robustness throughout training to estimate it. For every example, we average the probabilities obtained by the same 10 experiments introduced in Section C.1. As shown in Figure 11a, prediction probability is inversely correlated with data quality rank. A high-quality example is inclined to be correctly classified by the model with high probability. We do not estimate the data quality based on whether an example will be misclassified or not since the misclassification rate varies greatly across different training settings.

### D.2 MINIMUM PERTURBATION

Standard adversarial training often sets a perturbation radius universal to all training examples. However, it has been widely noticed that individual examples may have different levels of robustness against adversarial attacks. It might be helpful to customize the perturbation for each example during adversarial training. MMA (Ding et al., 2020) proposes a method to estimate proper individual perturbation for each example based on its distance to the decision boundary. The perturbation is determined by a line search along the perturbation direction initialized by a norm-constrained PGD attack. In a similar vein, IAAT (Balaji et al., 2019) performs a dynamic update of individual perturbation throughout the training. CAT (Cheng et al., 2020) further incorporates individual label smoothing based on the estimated perturbation level. Instead of customizing individual perturbation radius for each example, FAT (Zhang et al., 2020a) customizes the number of attack iterations for each example, such that the perturbation is just enough to fool the model. GAIRAT (Zhang et al., 2020b) further adopts a weighted loss function based on such individual attack iterations to focus

more on those examples far from the decision boundary. Nevertheless, none of these works detailedly analyzes the impact of such individual perturbation on the adversarial training.

Here, we show that profiling the data based on individual perturbation radius leads to similar result as that motivating from learning stability introduced in Section 3.1. We denote the *minimum perturbation* of an example as the smallest perturbation radius required to change a model's prediction on it. Ideally, the minimum perturbation of an example amounts to its minimum distance to the decision boundary. We use an untargeted attack based on I-FGSM (Iterative Fast Gradient Sign Method) (Goodfellow et al., 2015; Kurakin et al., 2017) with step size $1/255$ based on the implementation in Adversarial Robustness Toolbox (ART) (Nicolae et al., 2018). We empirically found the step size $1/255$ is often small enough to ensure a converged minimum perturbation. One can also employ other attacks including but not limited to DeepFool (Moosavi-Dezfooli et al., 2016), CW (Carlini & Wagner, 2017), EAD (Chen et al., 2018) and FAB (Croce & Hein, 2020a) in estimation. But note that all these methods cannot obtain the true minimum perturbation, but only the upper bound of it (Weng et al., 2018). Estimation of the minimum perturbation through optimization is known as a NP-hard problem (Katz et al., 2017).

In Figure 11b, we use the best model in terms of robustness obtained in training to estimate the minimum perturbation, and average the results obtained by the same 10 experiments mentioned in Section C.1. One can find that the minimum perturbation is inversely correlated with the data quality rank, which means that a low-quality example is more likely to reside near the decision boundary. This suggests the sophisticated methods mentioned above are likely to treat examples with different qualities differently. It also suggests that examples with different amounts of minimum perturbation will influence the adversarial training differently in terms of the contributions to robustness and aforementioned problems.

### D.3 LEARNING ORDER

Learning order refers to the phenomenon that Deep Neural Networks (DNNs) learn the examples in a similar order, which is shared by different random initializations and neural architectures. Such phenomenon is observed widely in standard training and training with noisy inputs and labels (Arpit et al., 2017; Li et al., 2020). It is demonstrated that the learning order originates from the coupling between DNNs and benchmark datasets (Hacohen et al., 2019), since DNNs learn synthetic datasets without a specific order and classifiers other than DNNs such as AdaBoost (Freund & Schapire, 1995) learn benchmark datasets without a specific order.

We show that learning order still exists in adversarial training. We denote the $1^{st}$ *learned epoch* as the first epoch when a training example is classified correctly under adversarial attack. We show there is a correlation between the $1^{st}$ learned epochs of an example across different training settings (see Section D.4). Furthermore, we show that learning order is correlated with data quality rank. In Figure 11c, we average the $1^{st}$ learned epochs of an example obtained by the same experiments as mentioned in Section C.1. One can find the $1^{st}$ learned epoch is positively correlated with its quality rank, which means low-quality examples are likely to be learned late during training.

Note that if we pick the best epoch[3] as a boundary and partition the training examples based on their $1^{st}$ learned epochs, the resulting two subsets correspond to exactly the correctly classified and misclassified examples. This implies that those adversarial training methods treating examples differently based on whether they are correctly classified or not, as mentioned in Section D.1, are likely to be a special case of treating examples differently based on their quality, from yet another perspective.

### D.4 CONSISTENCY OF THE MOTIVATION

We show that each motivation mentioned above is itself consistent across different training settings. Therefore it is possible to estimate the data quality rank similarly from each motivation. In Figure 12, 13, 14, we use the same experiments mentioned in Figure 2 to show that the prediction probability, minimum perturbation and learning order are all consistent across random initializations, different training methods, and neural architectures. Nevertheless, one may find that these estimations are

---

[3]The epoch when the best model in terms of robustness is obtained

relatively less consistent compared to the data quality rank estimated from learning stability, which is the major reason that we estimate the data quality rank from learning stability in the main text.

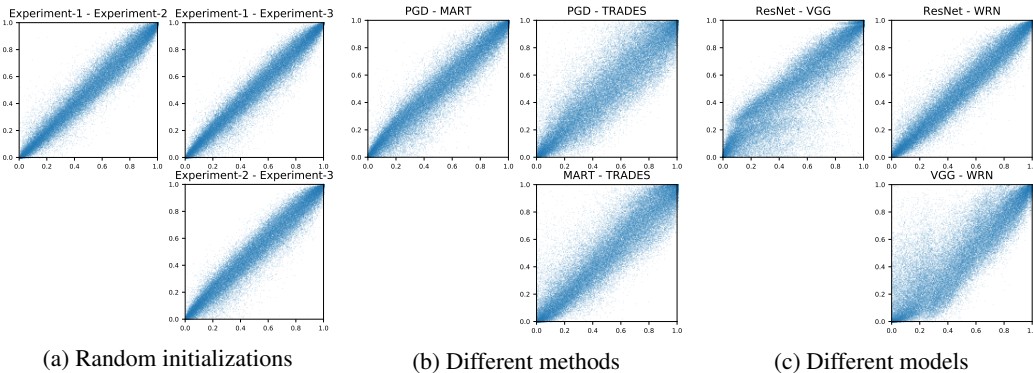

Figure 12: Scatter plots of the quality ranks of training examples based on prediction probabilities obtained by different training settings. The prediction probability of an example is consistent across random initializations, different methods and models.

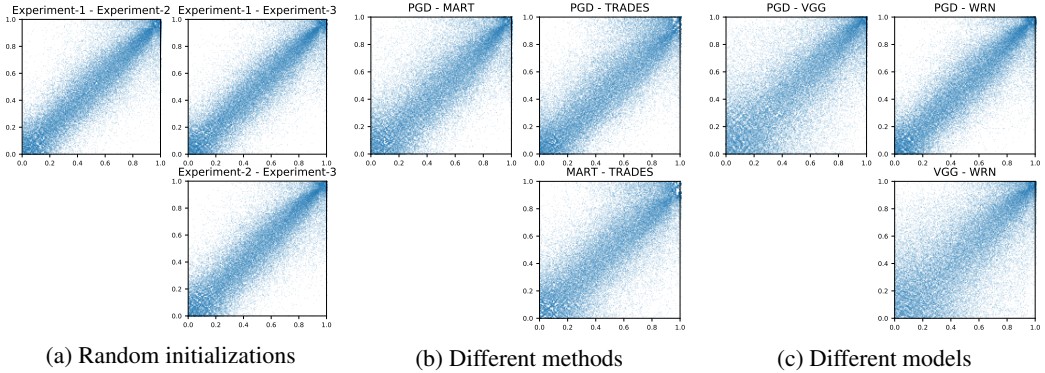

Figure 13: Scatter plots of the quality ranks of training examples based on minimum perturbations obtained by different training settings. The minimum perturbation of an example is consistent across random initializations, different methods and models.

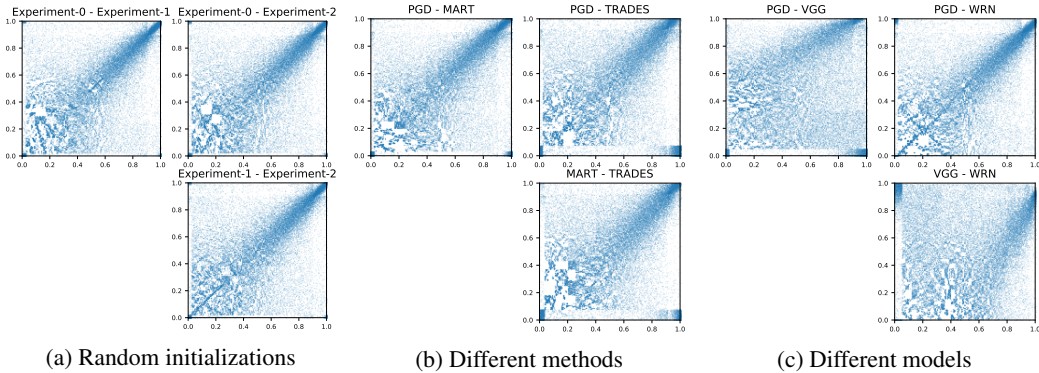

Figure 14: Scatter plots of the quality ranks of training examples based on 1st learned epochs obtained by different training settings. The 1st learned epoch refers to the first epoch when an example is classified correctly under adversarial attack, which is consistent across random initializations, different methods and models.

# E LIMITATIONS OF REMOVING LOW-QUALITY DATA

We note that removing low-quality training data cannot be advocated as a completing method. Although removing low-quality data can benefit the robustness in certain scenarios, it will inevitably impair the standard accuracy, since those hard examples are of high-quality to standard training. As shown in Table 3, removing the 20% examples with the lowest quality can improve the best robustness consistently for different training methods and perturbation radii, but will also diminish the standard accuracy for most settings (except PGD training with perturbation radius 12/255).

Table 3: Performance of adversarially training a pre-activation ResNet-18 on CIFAR-10 using PGD and TRADES with different training perturbation radii.

| Method | $\varepsilon$ (/255) | # Training samples removed (%) | Standard Acc (%) | Robust Acc (%) |
|---|---|---|---|---|
| PGD | 8 | 0 | 83.92 | 48.12 |
| PGD | 8 | 20% | 82.26 | 48.24 |
| PGD | 12 | 0 | 76.08 | 48.95 |
| PGD | 12 | 20% | 77.85 | 49.22 |
| TRADES | 8 | 0 | 81.73 | 48.18 |
| TRADES | 8 | 20% | 78.16 | 49.35 |
| TRADES | 12 | 0 | 76.56 | 45.40 |
| TRADES | 12 | 20% | 73.53 | 46.85 |

# F EXPERIMENT DETAILS

We adopt the following setting for all experiments unless otherwise noted.

**Robustness evaluation.** We consider the robustness against $\ell_\infty$ norm-bounded adversarial attack with perturbation radius $8/255$. Throughout the paper, the following attacks are employed.

- **AutoAttack**: This is currently the strongest adversarial attack to the best of our knowledge. No parameter setting is required.

- **PGD-10**: PGD with 10 attack iterations and step size fixed to $2/255$.

- **PGD-1000**: PGD with 1000 attack iterations. The step size is fixed to $2/255$, which is the best value suggested by Croce & Hein (2020b).

- **Square Attack:** This is a strong score-based black-box attack and is also built into the standard AutoAttack. We list it separately here to eliminate the possibility that the adaptive design in AutoAttack can be sensitive to the data. We adopt 1 restart with maximally 5000 queries.

- **RayS:** This is a strong decision-based black-box attack that only requires the final prediction of the model. We set the maximum number of queries to be 10000.

- **Transfer attack:** We use PGD-AT to adversarially train a WRN-28-10 on the CIFAR-10 training set. The model at the best checkpoint in terms of the robustness will then serve as a surrogate model to generate the adversarial attack.

**Adversary setting.** We conduct adversarial training with $\ell_\infty$ norm-bounded perturbations. We employ PGD-AT and TRADES as base methods, and two sophisticated variants GAIRAT and MART. We fix the perturbation radius to $8/255$ unless otherwise noted. The number of attack iterations is fixed as 10, and the perturbation step size is fixed as $2/255$. We adopt early stopping (Rice et al., 2020) as a default strategy and report the best robustness obtained throughout the training. Additional settings specific to each method are listed as follows.

- **TRADES:** The regularization hyperparameter is fixed as $1/\lambda = 6.0$ as recommended in the official implementation.

- **GAIRAT:** A sample-wise weighting based on the distance to the decision boundary is employed in GAIRAT. The distance to the decision boundary is approximated by the minimum number of iterations $\kappa(x, y)$ to generate adversarial perturbation that successfully fools the model in the inner maximization. The sample-wise weighting is given by

$$w(x, y) = \frac{1 + \tanh(\lambda + 5 \cdot (1 - 2 \cdot \kappa(x, y)/K))}{2},$$

  where $K$ is the maximum number of iterations. We fix $\lambda = 0$ throughout the training. $\omega(x, y)$ will be normalized to ensure $\sum_i^n \omega(x, y)/n = 1$, where $n$ is the total number of training examples.
- **MART:** We adopt the version that employ the boosted cross entropy (BCE) loss in the outer minimization. The regularization hyperparameter is fixed as $1/\lambda = 6.0$ to be aligned with TRADES.

**Training setting.** We employ SGD as the optimizer. The momentum and weight decay are set as 0.9 and 0.0005 respectively, which are aligned with the common practice. For Wide ResNet, we conduct the training for 120 epochs, with the learning rate starting at 0.1 and reduced by a factor 10 at epoch 100 and 110. For other neural architectures, we conduct the training for 160 epochs, with the learning rate starting at 0.1 and reduced by a factor of 10 at epoch 80 and 120.

**Dataset.** We conduct experiments on three datasets including CIFAR-10, CIFAR-100 and Tiny-ImageNet without additional data.

**Neural architecture.** We conduct experiments with pre-activation ResNet-18, VGG and Wide ResNet.

**Hardware.** We conduct experiments on NVIDIA Quadro RTX A6000..

## G EXPLANATION

In this section, we move one step further to probe into the effect of data quality in adversarial training. In previous works, it has been proven that standard classifier and robust classifier learn fundamentally different features (Tsipras et al., 2019). Useful features prevail in the dataset, but are not necessarily robust and comprehensible to human (Ilyas et al., 2019). In light of such analyses of robust/non-robust features, we motivate from a feature learning perspective and try to uncover the potential mechanisms of how low-quality data is interconnected with the existing difficulties in adversarial training including robustness-accuracy trade-off, robust overfitting and robustness overestimation. Under the assumption that similar features will be recognized in a specific class of examples, we selectively conduct the adversarial training on either one or a few classes of examples. In this way, we can isolate the impact of each set of similar features and analyze the effects of low-quality data on the learning of features.

### G.1 ROBUSTNESS-ACCURACY TRADE-OFF

Towards understanding the correlation between robustness-accuracy trade-off and data quality, we conjecture that low-quality data will cause the loss of useful features in adversarial training.

As shown in Figure 3a, low-quality examples are intrinsically ambiguous from a human perspective. Therefore, if the perturbation radius is relatively large, the adversarial attack may generate reasonable images of classes other than the true class. Indeed, as shown in 3a, the adversarial counterparts of low-quality examples catch salient characteristics of the classes that the model predicts. In contrast, the adversarial counterparts of high-quality examples are still explicit images of the true classes. Here, we adversarially attack an example using PGD-20 with perturbation radius $\varepsilon = 16/255$ based on the best model obtained through PGD-AT. The perturbation radius $16/255$ is exactly twice the perturbation radius commonly used in adversarial training, which implies that adversarial counterparts generated on the low-quality examples during the training might be marginal cases from a human perspective.

Based on the above observation, we suspect that the adversarial training might not be suitable for low-quality examples. It is explained in Ilyas et al. (2019) that adversarial training works because the adversary can exploit non-robust features of classes other than the true class, thus forcing the model

to rely only on the robust features of the true class. However, due to the ambiguity, such "distracting" features generated on a low-quality example might be too prominent such that it overwhelms the regular features of other classes when it is forced to be classified into the true class of this example, thus significantly damage the recognizability of other classes.

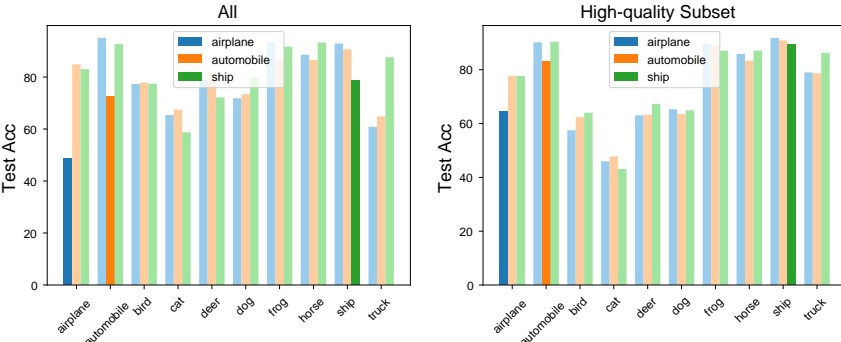

Figure 15: Fine tune the model obtained by standard training with adversarial examples generated by targeted attack on either all the training examples or the high-quality subset among them. When the attack in adversarial training is primarily targeted to "Airplane", the test accuracy of "Airplane" after fine-tuning is significantly lower, while the test accuracies of other classes are comparable. Instead, when the attack is targeted to "Automobile", the test accuracy of "Automobile" after fine-tuning is significantly lower. In contrast, when fine tuning the model only on the high-quality subset, such difference is not significant.

We manifest such loss of recognizability by robustly fine tuning the model obtained by standard training. Instead of untargeted attack, we use targeted attack in adversarial training[4] to isolate the effect of recognizability loss. Specifically, for the PGD attack employed in adversarial training, we replace the inner maximization by a minimization towards a target class $c$, except for those examples have class $c$ as their true labels, where the target is directed to another class $c''$. We refer this special case as $c - c''$ adversarial training. Figure 15 shows the average standard test accuracy produced by fine-tuning for 30 epochs using Airplane-Truck, Automobile-Truck and Ship-Truck adversarial training. These classes are selected because the model produces highest standard test accuracy on them. One can find that, when the attack in adversarial training is primarily targeted to "Airplane", the test accuracy of "Airplane" after fine-tuning is significantly lower, while the test accuracies of other classes are comparable. Instead, when the attack is targeted to "Automobile", the test accuracy of "Automobile" after fine-tuning is significantly lower. In contrast, when fine tuning the model only on the high-quality examples, such difference is not significant. This reflects the detail of how low-quality examples in adversarial training hurts the learning of useful features.

## G.2 ROBUST OVERFITTING

Here, we further show that the low-quality examples will also damage the robust recognizability. In adversarial training, the gradient-based adversary may generate robust features of another class on low-quality examples due to their intrinsic ambiguity, especially when the model is already relatively robust. Consequently, the model may forget the robust recognizability of other classes because it is forced to classify such robust feature to the original class, which leads to robust overfitting.

We manifest the loss of recognizability by fine-tuning the best model obtained by a regular adversarial training. Similarly, we use targeted attack to isolate the effect. Figure 16 shows the average robust test accuracy produced by fine-tuning the best model for 30 epochs using Airplane-Truck, Automobile-Truck and Ship-Truck adversarial training. These classes are selected because the model produces highest robust test accuracy on them. One can find similar results that, when the attack in adversarial training is primarily targeted to "Airplane", the robust test accuracy of "Airplane" after fine-tuning is significantly lower. Instead, when the attack is targeted to "Automobile", the robust test accuracy of "Automobile" after fine-tuning is significantly lower. In contrast, when fine tuning the model only on the high-quality data, such difference is not prominent.

---

[4]It is not common to use targeted attack in adversarial training, only for demonstration here.

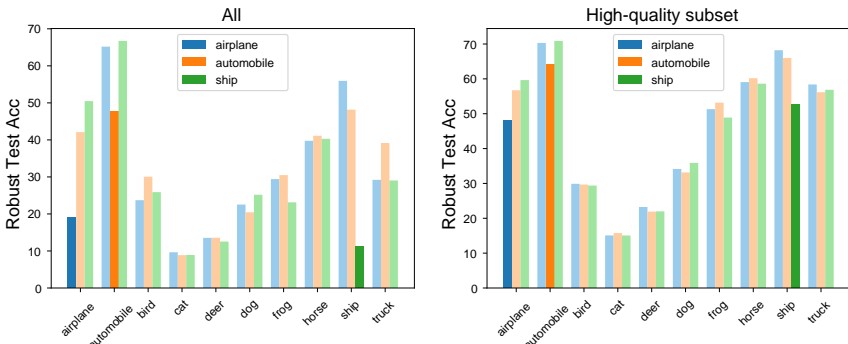

Figure 16: Fine tune the best model obtained in adversarial training with targeted attack on either all the training examples or the high-quality subset among them. When the attack in adversarial training is primarily targeted to "Airplane", the robust test accuracy of "Airplane" after fine-tuning is significantly lower, while the performance of other classes are comparable. We can observe similar patterns when the attack is targeted to "Automobile" or "ship". In contrast, when fine tuning the model only on the high-quality subset, the difference is insignificant.

## G.3 ROBUSTNESSS OVERESTIMATION

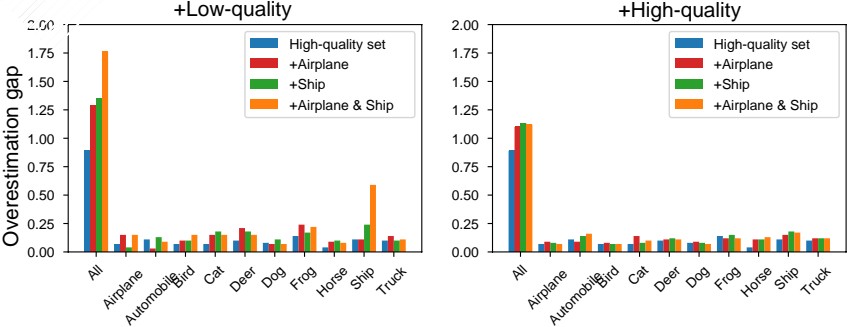

Figure 17: The overestimation gap, namely the difference between PGD-10 and Auto Attack evaluation, generated by adding examples of two competing classes into a high-quality subset.

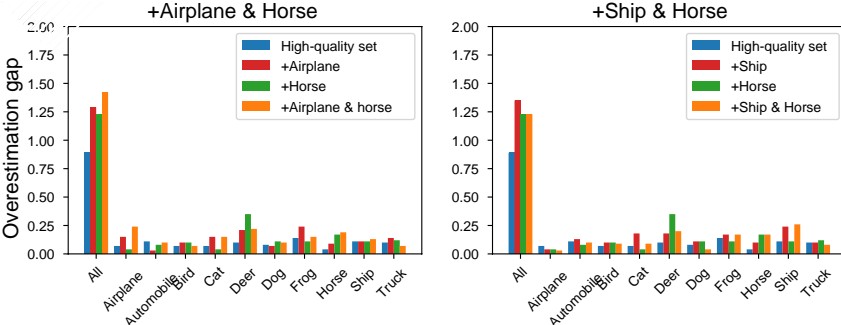

Figure 18: The overestimation gap generated by adding examples of two non-competing classes into a high-quality subset.

We show that the low-quality data causes robustness overestimation through a mechanism which we refer as "competing". In Figure 17, we sample a subset by adding 500 additional low-quality examples[5] to a class-balanced high-quality subset of size $10^4$, adversarially train the model on

---

[5] 10% of all the training examples in one class

it, and show the overestimation gap. One can find that when we only add examples of one class either "Airplane" or "Ship", the overestimation gap increases, but not significantly compared to the overestimation gap of the original high-quality subset. However, if we add the examples of two "competing" classes "Airplane" and "Ship" at the same time, the overestimation gap increases substantially, and mostly attributes to the class "Ship". Here, "competing" classes means these two classes contain images that are likely to have similar features[6]. The overestimation gap is not significant if the additional examples are not from two "competing" classes, as shown in Figure 18 where we add additional examples from either "Airplane" and "Horse", or "Ship" and "Horse". The overestimation gap is also not significant if the additional examples are not low-quality, as shown in Figure 17 where we add additional high-quality examples, even if they are from competing classes.

Towards understanding this mechanism, we focus on the inner maximization since overestimation mainly indicates weakened generation of adversarial perturbation, while previously we assume this process is ideal. Recall that adversarial attack works because in the inner maximization, the adversary optimizes towards classes other than the original class and thus can exploit distracting features to fool the model (Ilyas et al., 2019). However, since the ambiguous low-quality examples from different classes tend to contain similar features, the adversary may optimize towards different classes starting from similar features, subsequently damage its capability to exploit distracting features of these classes.

---

[6]One can refer to the sample images we showed in Figure 3a

