# OpenReview forum: "Data Quality Matters For Adversarial Training: An Empirical Study"
_ICLR.cc/2022/Conference — ICLR 2022 Submitted_

### Official Review · Reviewer_mNG6 · 2021-10-26

**Correctness:** 3
**Technical Novelty And Significance:** 2
**Empirical Novelty And Significance:** 4
**Recommendation:** 6
**Confidence:** 4

**Main Review:**

Strengths:
- Interesting direction- I find this work very exciting since it contributes to our understanding of adversarial training.  Many works have demonstrated that adversarial training requires more data.  This work expands along this direction by observing how the quality of data impacts generalization of adversarial training.
- Good scope in experiments, significant results- The authors perform experiments on multiple datasets and adversarial training methods and the trends look consistent across methods.  Patterns discovered between data quality and properties like robustness, robust overfitting, and robustness estimation also look significant.

Weaknesses:
- In the main text, it say that results for WideResNet architecture are in the appendix, but when I looked at the appendix I couldn't find figures for WRN corresponding to the experiments in the main text.  I think adding some figures for WRN with gradually removing lowest quality vs random would also help show the generalizability of the observed trends.
- In section 3.2 related works, the authors state that prior works show that more data can be detrimental to adversarial training, but I don't agree that is the case.  I think prior works demonstrate that adversarial training benefits a lot from additional data.  For instance, Gowal et al. 2020 is cited to support the statement that additional data unrelated to the original dataset can hurt adversarial robustness, but Gowal et al. demonstrate that additional data always improves over the no additional data case.  There is also discussion of label noise in this section, but it isn't clear how label noise is connected to the idea of additional data and data quality.  Additionally, I think discussion of other works which measure notions of hardness and data quality for adversarial training can also be added to related works.  For instance, methods of measuring difficulty of adversarial examples used for curriculum adversarial training [1,2,3].  Recently there has also been a line of works discussing ways to measure the data quality of additional data generated by generative models for use with adversarial training [4, 5].


[1] Wang, Yisen, et al. "On the Convergence and Robustness of Adversarial Training." ICML. Vol. 1. 2019.
[2] Zhang, Jingfeng, et al. "Attacks which do not kill training make adversarial learning stronger." International Conference on Machine Learning. PMLR, 2020.
[3] Cai, Qi-Zhi, Chang Liu, and Dawn Song. "Curriculum adversarial training." Proceedings of the 27th International Joint Conference on Artificial Intelligence. 2018.
[4] Sehwag, Vikash, et al. "Robust Learning Meets Generative Models: Can Proxy Distributions Improve Adversarial Robustness?" arXiv preprint arXiv:2104.09425 (2021)
[5] Gowal, Sven, et al. "Improving Robustness using Generated Data." arXiv preprint arXiv:2110.09468 (2021).

**Summary Of The Paper:**

The paper investigates the impact of data quality, measured by the proportion of epochs in which the model classifies a specific input correctly during training, on the robustness, generalization, and robustness-accuracy tradeoff of adversarially trained models.  Unlike with standard training, the authors find that more difficult inputs (lower quality inputs) can hurt adversarially trained models.  They find that compared to randomly removing data during training, removing low quality data can lead to higher robustness, less robust overfitting, less robustness overestimation, and less robustness-accuracy tradeoff.

**Summary Of The Review:**

I vote to accept this paper since I find the empirical contributions significant; the authors rigorously study the impact of data quality on various properties of adversarially trained models including robustness, robust overfitting and robustness overestimation, and robustness-accuracy tradeoff.  I think the discussion of related works can be improved though.

---

> ### Author Response · Authors · 2021-11-23
> **Response to Reviewer mNG6**
>
> We thank the reviewer for the valuable comments. Please check our responses below.
>
> ## ____
> **Results on Wide ResNet**
>
> We include the experiment results on Wide ResNet showing the effect of removing low-quality examples on the robust test accuracy in Appendix B.2 (Appendix B.1 in the previous submission), but forget the results showing other problems. We apologize for this and have added the results on WRN-28-10 showing the correlation between data quality and three problems in adversarial training, including robust overfitting, robustness overestimation, and robustness-accuracy trade-off. Please check Appendix B.1 in the updated submission for the new figures and discussion. We sincerely thank the reviewer for the reminder.
>
> ## ____
> **Related work in Section 3.2**
>
> First, we agree with the reviewer that the statement “more data will not always yield better performance” in Section 3.2 is indeed confusing as more data indeed helps in cases such as additional data is added. A more rigorous statement should be “sorely increasing the sample size will not always yield better performance”. Therefore we also need to take data quality into consideration.
>
> Second, the discussion of label noise in this section aims to exclude the possibility that those low-quality data examples are simply label noise, in which case they cannot benefit the performance simply because their labels are incorrect. We have revised the relevant statements in Section 3.2 of the submission.
>
> ## ____
> **Additional related work on measuring data quality**
>
> We thank the reviewer for bringing these interesting works. We have added the discussion of these additional works to Section 3 in the updated submission. We note that we also have an independent section in the Appendix discussing different measures of data quality for adversarial training (Section D) and standard training (Section A.3).

---

> > ### Comment · Reviewer_mNG6 · 2021-11-29
> > **Post Rebuttal**
> >
> > Thank you for the response.

---

### Official Review · Reviewer_soLM · 2021-10-29

**Correctness:** 3
**Technical Novelty And Significance:** 2
**Empirical Novelty And Significance:** 2
**Recommendation:** 6
**Confidence:** 4

**Main Review:**

Strengths:

1.	This paper evaluates the learning stability of instances under various settings, which indicates that the estimation of quality for each data is almost accurate and consistent.

2.	This paper is well organized. The authors analyze the influences of low-quality data on the three issues separately, which empirically demonstrates removing the low-quality data can mitigate the issues.

Weakness:

1.	Calculating the quality of data is time-consuming and computationally expensive. The quality of data is closely related to a particular dataset. That is to say, when the dataset is updated such as adding some new data into the dataset, the quality of data in the new dataset needs to be recalculated. It should be a limitation for the practical use of low-quality data.

2.	Removing the low-quality data sometimes marginally improves adversarial robustness, but more often hurts the adversarial robustness and natural generalization. Therefore, the significance of the proposed low-quality data seems minor.


**Summary Of The Paper:**

This paper proposes a metric for evaluating the learning stability of data and point out that unstably-learned instances are of low-quality for adversarial training. Through extensive controlled experiments, this paper investigates the impact of low-quality data on three issues in adversarial training, i.e., robust overfitting, robustness overestimation, and robustness-accuracy trade-off. The experimental results show that removing the low-quality instances can mitigate the issues.

**Summary Of The Review:**

Overall, this paper systematically study the influences of data quality on the three problems in adversarial training. However, I am skeptical about the practical values of the proposed metric for data quality.

---

> ### Author Response · Authors · 2021-11-23
> **Response to Reviewer soLM**
>
> We thank the reviewer for the valuable comments. Please check our responses below.
>
> ## ____
> **Learning stability as a measure for data quality**
>
> First, we wish to note that our focus in this paper is to study the influence of data quality on problems in adversarial training, rather than to propose a better measure for data quality. Multiple other measures for data quality such as probability score and minimum perturbation have been discussed in Appendix D, which lead to consistent conclusions.
>
> Second, we note that learning stability has many potential applications. We list a few as follows.
> * Learning stability can be employed to select high-quality data in external or generated datasets with or without labels. The probability score is usually employed in the literature to select additional data [1, 2, 3]. However, we note that the output probability of a neural network is known to be poorly calibrated [4] while learning stability is likely to be more consistent in terms of ranking based on our experiments.
> * Learning stability can be employed to customize the inner maximization or outer maximization in adversarial training on the fly for every example. Typical strategies existing in the literature include reweighting the examples in the loss function [5], selecting appropriate attack iterations for different examples [6], etc. All these strategies can be easily adapted based on learning stability and be potentially improved. We note that calculating learning stability during training doesn’t require any additional computation load as the model has to make predictions on the training examples anyway.
> * Learning stability can be employed to perform curriculum learning/self-adaptive learning, namely prioritizing the learning of a portion of examples during training. Existing techniques to identify preferred examples including convergence quality [7], etc., which can be easily adapted based on learning stability. Again, we note that no additional computation is required to calculate learning stability in this case as discussed above.
>
> [1] Unlabeled Data Improves Adversarial Robustness. Carmon et al., 2019.\
> [2] Uncovering The Limits Of Adversarial Training Against Norm-Bounded Adversarial Examples. Gowal et al., 2020.\
> [3] Improving adversarial robustness using proxy distributions. Sehwag et al., 2021.\
> [4] On Calibration of Modern Neural Networks. Guo et al., 2017.\
> [5] Geometry-Aware Instance-Reweighted Adversarial Training. Zhang et al., 2021.\
> [6] Attacks Which Do Not Kill Training Make Adversarial Learning Stronger. Zhang et al., 2020.\
> [7] On the Convergence and Robustness of Adversarial Training. Wang et al., 2019.

---

> > ### Comment · Reviewer_soLM · 2021-11-29
> > **Thank you for the rebuttal**
> >
> > I thank the authors for the detailed response. I agree that the contribution of empirically disclosing the influence of data quality on problems in adversarial training is good.
> > In addition, learning stability can be applied to various applications and using learning stability in a on-the-fly manner does not incur extra computation. However, I am concerned about whether applying learning stability to adversarial training can achieve a superior performance compared to baselines such as AT with unlabled data [1], GAIRAT [2], FAT [3], DAT [4]. The experiments in the main paper (Figure4-6) are conducted based on the learning stability that is calculated and determined before training. It could be better if some superior experimental results of using learning stability on-the-fly in adversarial training are shown in the paper. Overall, I increase my score to 6.
> >
> > [1] Unlabeled Data Improves Adversarial Robustness. Carmon et al., 2019.
> >
> > [2] Geometry-Aware Instance-Reweighted Adversarial Training. Zhang et al., 2021.
> >
> > [3] Attacks Which Do Not Kill Training Make Adversarial Learning Stronger. Zhang et al., 2020.
> >
> > [4] On the Convergence and Robustness of Adversarial Training. Wang et al., 2019.

---

> > > ### Author Response · Authors · 2021-11-30
> > > **Thank you for the reply**
> > >
> > > We thank the reviewer for the additional comments and for increasing the score. We will follow the valuable suggestions and include more results in the paper. Could you please update the score in the review system to reflect the increase?

---

### Official Review · Reviewer_Rp8o · 2021-11-02

**Correctness:** 4
**Technical Novelty And Significance:** 3
**Empirical Novelty And Significance:** 3
**Recommendation:** 6
**Confidence:** 5

**Main Review:**

**Strengths**:
* The paper has a clear message that adversarial training and nominal training do not benefit from the same type of samples.
* Extensive experiments on CIFAR-10 to analyze the robust performance of "high quality" data (selected by their proposed criterion) under three angles commonly discussed in the Lp-norm community.
* The paper is well written and very clear.

**Weaknesses/Suggestions/Questions**:
1) **Main Weakness**. It really lacks of an analysis in the extra data case. This case should be actually perfectly suited for the message of the paper as extra data for CIFAR-10 is known to be of lesser quality (as per the references in the paper). Similarly, recent works [1, 2] have proposed to use generated data in the Lp-norm case. The proposed quality criterion in the paper would be great to analyze the quality of the generated images compared to the original CIFAR-10 images.
2) In Figure 3, "with their labels predicted by a model", which model? This is a bit vague.
3) Figures 4 to 7. First, please put in the caption how the robust test accuracy is obtained. Against PGD-10, PGD-40? It is unclear. Second, all the first rows are labeled "PGD". I think you mean "Adversarial Training" as introduced by [3]. PGD is the optimization procedure and it also used on TRADES. So the naming "PGD" is misleading.
4) Typo: "can also be found in our Figure 1(a)" -> "can also be found in our Figure 1(c)" on page 3.

**References**:
1) Sylvestre-Alvise Rebuffi, Sven Gowal, Dan A Calian, Florian Stimberg, Olivia Wiles, Timothy Mann. *Fixing data augmentation to improve adversarial robustness*. 2021.
2) Vikash Sehwag, Saeed Mahloujifar, Tinashe Handina, Sihui Dai, Chong Xiang, Mung Chiang, Prateek Mittal. *Improving adversarial robustness using proxy distributions*. 2021.
3) Aleksander Madry, Aleksandar Makelov, Ludwig Schmidt, Dimitris Tsipras, Adrian Vladu. *Towards deep learning models resistant to adversarial attacks*. 2017.

**Summary Of The Paper:**

**Few sentences summary**: the paper proposes an empirical study on how data quality impacts robust performance in the Lp-norm setting under three angles: robust overfitting (i.e. the difference in accuracy between the best and last checkpoints), robustness evaluation (i.e. consistency of the robust performance when trying various robust evaluations like AutoAttack) and clean/robust accuracy trade-off (i.e. the gap in performance between the clean and robust accuracies). They show that "high quality" data (selected with their proposed criterion) performs better than "low quality" data for these three performance metrics.

Regarding the **results** and **contributions**:
* The authors propose a quantitative definition of "quality"  based on the average training robust accuracy over epochs per sample.
* They show that "low quality" samples are better for standard training whereas "high quality" samples are better for adversarial training.
* Studies on 1) robust overfitting, 2) robustness evaluation and 3) clean/robust accuracy trade-off on the datasets CIFAR-10/100 and TinyImageNet for Adversarial Training and TRADES.

**Summary Of The Review:**

Empirical study paper with an interesting message that adversarial training and nominal training do not benefit from the same type of samples. The paper could have gone more in depth by studying the extra data case which is commonly used in the literature and gets the SOTA results. Especially, this extra data case normally fits the message of the paper so it is a missing element in the paper.

---

> ### Author Response · Authors · 2021-11-23
> **Response to Reviewer Rp8o**
>
> We thank the reviewer for the valuable comments. Please check our responses below.
>
> ## ____
> **Extra data is an exciting future work opportunity for our work**
>
> We agree with the reviewer that it is important to study the effectiveness of different data quality measures, especially for the extra data case. However, the message we wish to deliver in this study is that data quality has an important effect on various problems in adversarial training. We have shown in Appendix D that multiple data quality measures support our claim. We select learning stability as an example to measure the data quality mainly because it produces a more stable result across different experiment settings. It is really not our focus to analyze the effectiveness of these different measures on applications such as selecting extra data. Which measure is more favorable in selecting extra data and how to design a better quality measure to excel in the performance is a complicated problem that goes beyond the scope of this study and deserves an independent study, especially in considerations of multiple aspects including effectiveness, efficiency, fairness, etc.
>
> Nevertheless, we would like to thank the reviewer for bringing the recent works [1] and [2] that we were previously unaware of. We have listed the analyses of data quality measures for extra data as an important future work.
>
> ## ____
> **Clarification of experiment details**
>
> * Figure 3: Here the labels are predicted by a pre-activation ResNet-18 adversarially trained on CIFAR-10 with PGD-AT [3]. Detailed training settings can be found in Appendix F.
> * Evaluation of robust test accuracy in Figures 4, 5, 6: We employ AutoAttack to evaluate the robust accuracy by default. To show the robustness overestimation in Figure 6 we also employ PGD-10 attack, with perturbation radius 8/255 and step size 2/255.
> * “PGD” in Figures 4, 5, 6,7: The column title “PGD” means PGD training. We apologize for the confusion and have used PGD-AT to refer to adversarial training with PGD in the submission.
> * Typo on page 3: We thank the reviewer for pointing out this typo and have updated it in the submission.
>
> [1] Fixing data augmentation to improve adversarial robustness. Rebuffi et al., 2021.\
> [2] Improving adversarial robustness using proxy distributions. Sehwag et al., 2021.\
> [3] Towards deep learning models resistant to adversarial attacks. Madry et al., 2017.

---

### Official Review · Reviewer_VxQb · 2021-11-16

**Correctness:** 3
**Technical Novelty And Significance:** 2
**Empirical Novelty And Significance:** 2
**Recommendation:** 6
**Confidence:** 3

**Main Review:**

(Edited score post rebuttal)

Strengths:
(i) This paper studies an important problem on the effect of data quality on different aspects of robust training.
(ii) The paper starts with the intuitive idea that data quality should matter and attempts to make a systematic empirical investigation across multiple datasets.
(iii) Overall, the paper is easy to read and the experiment details/results are clearly presented.
(iv) The observation that different sets of points improve robust vs standard accuracy is quite interesting to me (though that wasn't the focus of the paper).

Weaknesses: While the paper makes an interesting investigation, I worry that the conclusions/implications of the study are unclear and some inferences seem problematic to draw.
(i) One claim the paper makes is that low quality data can be detrimental to robust accuracy. Concretely, if we remove points from the dataset, that would improve robustness. Looking at Figure 4, the cases where this happens are: CIFAR-100 with TRADES and Tiny Imagenet with TRADES. I am confused why the conclusions vary between TRADES and Adversarial training? How is the \beta parameter set in the TRADES experiments. The fact that TRADES and adversarial training differ in their trends suggests that the beta parameter that controls accuracy vs robustness is important. The paper mentions that the hardness is an "intrinsic" measure since the relative ordering doesn't vary much across methods. So is it true that removing the *same* points leads to different performance with TRADES and adversarial training?
(ii) Other than the (slightly questionable) claim that removing low quality data improves performance, I didn't find the other implications on robustness particularly surprising. Even in standard training, it has been shown that there are some "support vectors" (or prototypes) which effectively dictate the final model learnt, and the rest of the dataset does not matter as much. The paper seems to mainly replicate this for robust training.
(iii) Computing the data quality measure: If I understand correctly, you need to compute the robust accuracy across the training checkpoints. Which attack do you use to compute this? Does it matter how strong the attack is?
(iv) What's an actionable takeaway from this work? Aside from the caveat of point (i) above on whether there is a reliable takeaway, I wonder how to actually operationalize the idea of using higher quality data? The data quality of each point seems to depend on the remaining points (since it comes from some ordering of robust accuracy), and computing this requires training the model on all the data. Can the authors please comment with any thoughts here?
(iv) Computing data quality for extra data: Can you use checkpoints trained on CIFAR-10 to evaluate data quality of additional data (for e.g. from Carmon et al. 2019) not present in the training set? Basically, this would involve computing the robustness at the new point (not in training) across different training epochs. If the authors could show that this improves performance over adding all the mined extra points, I would be convinced that there is something interesting here. Relatedly, how does this measure of data quality empirically differ from just logits for confidence (which was what was used to mine the extra data in Carmon et al. 2019).


**Summary Of The Paper:**

This paper studies the effect of data quality on adversarial robustness. Specifically, they focus on one measure of data quality (number of times there is a perturbation that is misclassified across training iterations). They study the effect of data quality on robust overfitting, robustness-accuracy tradeoffs and "robustness overestimation" (gap between strong and weak attacks). The main conclusions reported are that data quality as measured by their metric plays an important role in all three aspects, and a suggested takeaway is that we need data of higher quality to improve robustness.

**Summary Of The Review:**

This paper studies an interesting question about the effect of data quality on robustness. Unfortunately, I find the conclusions for robustness as emphasized in the paper either unsubstantiated (removing low quality data improves robustness) or difficult to make actionable. Nevertheless, there are some interesting empirical observations in the paper such as the difference between important points for standard and robust accuracy, the effect of data quality on robustness-accuracy tradeoffs which were not known previously. I apologize for the late review - there was some issue at my end and I realized the review didn't get submitted. I would really appreciate if the authors could respond to my questions above. Thanks!

---

> ### Author Response · Authors · 2021-11-23
> **Response to Reviewer VxQb**
>
> Thanks for your valuable and timely comments! We have tried our best to prepare thorough answers to your questions.
>
> ## ____
> **Removing low-quality data as a method is not our focus**
>
> We would like to emphasize that improving the performance by removing low-quality data is not our focus and should not be considered as a practical approach since (1) it may hurt the clean accuracy (2) it raises fairness issues. For the experiment details, we note that TRADES and PGD-AT can indeed have different behaviors when removing low-quality data, which may be due to the fact that TRADES can underfit low-quality data as discussed in Paragraph 3 of Section 5.
>
> ## ____
> **Differences of our analyses from the observation in standard training**
>
> We sincerely believe our analyses of data quality in adversarial training are novel and inspiring. In specific, although there are data examples that do not contribute to the final performance (or the “support vectors” mentioned by the reviewer) in standard training, we note that they are easy examples [1]. This is in sharp contrast to our observation in adversarial training, where hard-to-learn or low-quality examples do not contribute to the performance. We believe this observation provides new insights for understanding the difficulties of robust learning.
>
> Most importantly, we note that our analyses extend to intriguing problems in adversarial training that are not seen in standard training, including robust overfitting, robustness overestimation, and robustness-accuracy trade-off. The analyses of data quality shed new insights to understand and solve those puzzles in adversarial training. In specific,
> * **Low-quality data induces robust overfitting**: Robust overfitting is a behavior that exhibits during training. Since simply removing the low-quality data mitigates this issue, it implies a fundamental connection between data quality and overfitting in adversarial training. In fact, recent work motivated by this observation has shown that low-quality data may result in implicit label noise [2], leading to robust overfitting.
> * **Low-quality data induces robustness overestimation**: Reliable evaluation of robustness in adversarial training is a notoriously difficult problem. Our data quality perspective is among the first that analyzes the origin of this problem. We believe the observation that removing low-quality data alleviates this issue cannot be simply explained by the existence of “support vectors”.
> * **Correlation between data quality and robustness-accuracy trade-off**: Robustness-accuracy trade-off is a well-known problem for robust learning. The observation that low-quality data induces the trade-off between robustness and accuracy implies robust and standard learning may rely on fundamentally different features. We believe this perspective opens new opportunities to understand this problem.
>
> Finally, we would like to note that the low-quality data are defined consistently throughout the paper, which means that all these problems in adversarial training can be analyzed and alleviated from this single perspective. We believe this is the first time that all these problems are able to be studied in a holistic manner, which should be a substantial contribution.
>
> ## ____
> **No additional attack is required to compute the learning stability for the data quality measurement**
>
> Learning stability is essentially the number of epochs that a data example will be predicted correctly under adversarial attack during training. We note that in adversarial training, the adversarial example of a training example will be generated in the inner maximization anyway. Therefore we can simply get the model prediction on such adversarial examples, and no additional attack is required.

---

> > ### Author Response · Authors · 2021-11-23
> > **Response to Reviewer VxQb (continued)**
> >
> > ## ____
> > **Practical implications**
> >
> > We note that this study offers a new perspective to investigate and potentially solve the problems in adversarial training in a holistic manner. We believe it opens new possibilities to many applications. Here we name a few for reference.
> >
> > Mitigate the problems in adversarial training from a data quality view:
> >
> > * **Robust overfitting**: The observation that low-quality data induces robust overfitting suggests that to mitigate robust overfitting we have to treat data with different quality levels differently in adversarial training. In fact, recent work has successfully mitigated robust overfitting with knowledge distillation [3], which can be viewed as an instance-wise label smoothing [4] that reduces the impact of low-quality data since their predictive probabilities are of high entropy.
> > * **Robustness overestimation**: The observation that low-quality data induces robustness overestimation suggests that emphasizing low-quality data in adversarial training can produce spuriously high robustness. We have discussed two adversarial training methods [5, 6] in Section 4.2 that may suffer from this issue. We note that this offers practical guidance for future works to design reliable adversarial training methods.
> > * **Robustness-accuracy trade-off**: The observation that robustness-accuracy trade-off is not significant on high-quality data mainly offers a new perspective to understand the robustness-accuracy trade-off. Nevertheless, practical suggestions are also promising such as how to properly deal with low-quality data to alleviate the trade-off.
> >
> > Improve the performance from a data quality view:
> > * The observation that low-quality data contributes differently to adversarial robustness and standard accuracy suggests that we need to customize the inner maximization (e.g. the perturbation radius, the number of iterations) and/or outer maximization (e.g. the loss function, regularization) for every example in adversarial training. Common strategies including instance reweighting, instance-wise attack strength, etc. We note TRADES is a successful pioneer work in this vein. The loss function of TRADES consists of a cross-entropy term that contributing to the clean accuracy, and a KL divergence term penalizing the difference between clean predictions and perturbed predictions, which contributes to the robustness. Since low-quality data examples have high-entropy predictive probabilities, TRADES will leverage the low-quality data for more clean accuracy instead of robustness, thus alleviating the trade-off. Sophisticated methods can be designed using better measures of data quality (e.g. learning stability) to go further in this direction.
> >
> > * Another way to customize the training for individual examples is curriculum learning. Since low-quality examples are often more ambiguous, the model may have to learn sufficient robust features from high-quality examples before it can learn features from low-quality examples. Previous works have demonstrated the effectiveness of prioritizing the learning of specific examples [7], which can be further improved by using better data quality measures.
> >
> > ## ____
> > **Extra data is an exciting future work opportunity for our work**
> >
> > We agree with the reviewer that extra data is an exciting follow-up opportunity for our work. As we mentioned in the related work section in the main paper, selecting high-quality extra data can indeed improve the robustness compared to adding all the extra data. In specific, [8] show that selecting 500k high-quality data can get better robustness than selecting 1M data from 80M-Ti, which is the dataset used in [9] to provide additional unlabeled data.
> >
> > In our submission, we aim to deliver a comprehensive, focused study on the in-distribution data first. We sincerely believe that determining which quality measure is better to select extra data and how to design a better quality measure is beyond the scope of our study and deserve an independent and in-depth study, in consideration of the effectiveness, efficiency, fairness, etc.
> >
> > [1] An Empirical Study Of Example Forgetting During Deep Neural Network Learning. Toneva et al., 2019.\
> > [2] Double Descent In Adversarial Training: An Implicit Label Noise Perspective. Dong et al., 2021.\
> > [3] Robust Overfitting May Be Mitigated By Properly Learned Smoothening. Chen et al., 2021.\
> > [4] Self-Distillation as Instance-Specific Label Smoothing. Zhang et al., 2020.\
> > [5] Geometry-Aware Instance-Reweighted Adversarial Training. Zhang et al., 2021.\
> > [6] Improving Adversarial Robustness Requires Revisiting Misclassified Examples. Wang et al., 2020.\
> > [7] On the Convergence and Robustness of Adversarial Training. Wang et al., 2019.\
> > [8] Uncovering The Limits Of Adversarial Training Against Norm-Bounded Adversarial Examples. Gowal et al., 2020.\
> > [9] Unlabeled Data Improves Adversarial Robustness. Carmon et al., 2019.

---

> > > ### Comment · Reviewer_VxQb · 2021-11-29
> > > **Post rebuttal**
> > >
> > > I thank the authors for a very detailed response to all my questions. I will update my score to 6 - I think the empirical results are interesting enough to merit further investigation and discussion by the community.
> > >
> > > I am still mildly skeptical of the universality of the notion of "data quality" as measured in the paper because different training methods behave differently and the data quality measured in the paper is tailored to a particular training algorithm. I also think that if atleast one of the practical implications were systematically investigated and shown to hold, I'd score an 8 on this paper.

---

### Decision · Program_Chairs · 2022-01-20

**Decision:**

Reject

**Comment:**

This paper studies the effect of data quality on adversarial robustness. It focuses on a single measure of data quality (number of times there is a perturbation that is misclassified across training iterations). The authors study the effect of data quality on robust overfitting, robustness-accuracy tradeoffs and "robustness overestimation" (gap between strong and weak attacks). The main conclusions reported are that data quality as measured by their metric plays an important role in all three aspects, and a takeaway is that data of higher quality may improve robustness. While the reviewers appreciated the premise of this work, some concerns remain post rebuttal. For example, few reviewers remain skeptical of the universality of the notion of "data quality" as measured in the paper because different training methods behave differently and the data quality measured in the paper is tailored to a particular training algorithm. Some reviewers also opined that at least one of the practical implications discussed in the rebuttal should be systematically investigated and that it is important to study the effectiveness of different data quality measures, especially for the extra data. Given all this, we are unable to recommend acceptance at this time. We hope the authors find the reviewer feedback helpful.